# Diffusion Models Preferentially Memorize Prototypical Examples
# or: Why Does My Diffusion Model Love Slop?

**Marta Aparicio Rodriguez** [1]   **Anastasia Borovykh** [1] [2]   **Grigorios A. Pavliotis** [1]   **Daniel J. Korchinski** [3]

## Abstract

Generative models have a persistent limitation: their tendency to memorize training data can create legal liabilities and erode creative diversity. Understanding which samples are memorized in whole or in part, and under what conditions, therefore remains an important open problem. Here we answer the question "Are atypical or rare samples memorized first?" in the negative. We train diffusion models on strings generated according to the production rules of the Random Hierarchy Model (RHM), and find that samples composed of *common substrings* are preferentially memorized. This holds true even if the training data consists of entirely unique samples, indicating that deduplication at the data point level does not provide a meaningful privacy guarantee. Correspondingly we predict, then observe, delayed memorization for fat-tailed datasets (i.e., those with more atypical samples). This effect is amplified when fat-tails are introduced into high-level production rules. These together suggest that *dataset diversity*, particularly at higher levels of abstraction, plays an important role in staving off memorization. Finally, we identify an intermediate regime of partial memorization in which common substrings are learned first and subsequently overproduced during generation. If training is stopped in this regime, models will exhibit the reversion-to-the-mean blandness often derided as "slop".

## 1. Introduction

Generative models are trained with the objective of creating new samples that are consistent with, but not identical to the training data. Over the past years, we have observed incredible success in the generation of different data modalities including text (OpenAI, 2022), image (Rombach et al., 2022; Ramesh et al., 2022), and video (Ho et al., 2022; Brooks et al., 2024). The creative ability of models to compose new data and expand beyond the training set (Kamb & Ganguli, 2025; Favero et al., 2025a) will determine whether AI becomes a genuine creative partner in our daily lives or remains perpetually dependent on human-generated data (von Werra, 2025; Foody, 2025). As compared with human output, language models produce text that overuses certain clichés (e.g., "It's not X, it's Y") or syntactic forms (Shaib et al., 2024) observed in training; we take such distribution shifts towards common motifs as definitional of the often derided *AI slop*. Complementary to the question of creativity is that of memorization: the degree to which models reproduce, rather than combine, their training data. In practice, memorization has been linked to a range of practical concerns, including copyright and privacy risks or the reproduction of biases present in data (Carlini et al., 2021; Bender et al., 2021; Carlini et al., 2023a). These issues can undermine trust in generative models and limit their safe and reliable deployment.

In this work, we explore the mechanisms of memorization and in particular seek to resolve competing hypotheses (see first row of Figure 1) that have appeared in the literature on how and when memorization arises.

**Hypothesis 1: outliers are memorized first** Once a model has largely achieved generalization, the remaining training loss is dominated by atypical or rare samples. These outliers therefore contribute disproportionately to the loss and, consequently, to the gradient signal used to update the model parameters. Such dynamics have been documented in classification models, where generalization is followed by the memorization of outlier or mislabeled examples (Feldman & Zhang, 2020; Feng & Tu, 2021). In the context of diffusion models, Pham et al. leveraged the viewpoint of energy-based models to show densely clustered data points can give rise to stable attractors that do not correspond to individual training samples (spurious states), while the isolated data points induce more distinct basins of attraction, and would therefore be easier to memorize.

[1]Department of Mathematics, Imperial College London, UK [2]ML Lab, Capital Fund Management, France [3]Department of Physics, École Polytechnique Fédérale de Lausanne (EPFL), Switzerland. Correspondence to: Marta Aparicio Rodriguez <marta.aparicio-rodriguez22@imperial.ac.uk>.

*Proceedings of the 43rd International Conference on Machine Learning*, Seoul, South Korea. PMLR 306, 2026. Copyright 2026 by the author(s).

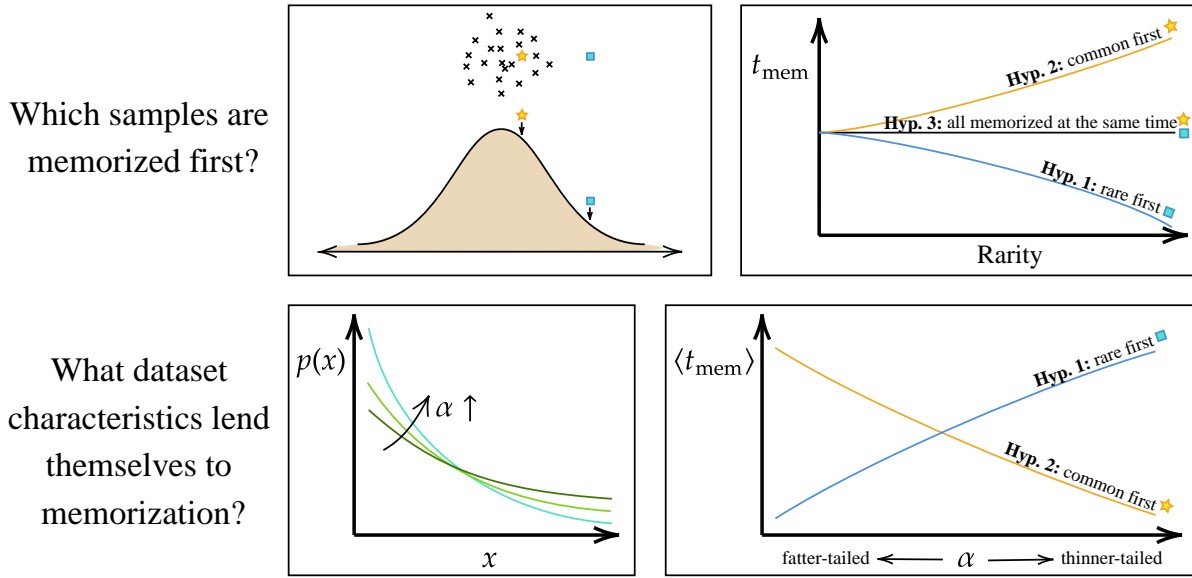

*Figure 1.* **Are rare data memorized first or last?** Data are drawn from distributions, and some data are outliers. For generative models, are such data memorized first? *Top left:* a data distribution. *Top right:* competing hypotheses for the effect of rarity on a given datum's memorization time. *Bottom left:* distributions with tails characterized by exponent $\alpha$. *Bottom right:* interaction of distribution rarity and time to memorization.

**Hypothesis 2: data with common features are memorized first**   Prior work has shown that duplicated data points tend to be memorized earlier during training (Lee et al., 2022; Carlini et al., 2023b; Aerni et al., 2025), likely because repeated exposure reinforces consistent gradient signals during training. At a more fine grained level, frequently occurring features may similarly be memorized more rapidly due to their consistent contribution to the training objective. This could, in turn, reduce the time required to reproduce samples composed predominantly of such features, compared to samples whose features are less common.

**Hypothesis 3: rarity of samples is irrelevant to memorization time**   Much of the literature in generative models treats memorization as a dataset-level phenomenon characterized by a global training time. Rather than analyzing when individual samples are memorized, these works identify a critical number of training steps, dataset passes or samples seen, beyond which models begin to reproduce data (Tirumala et al., 2022; Gu et al., 2025a; Bonnaire et al., 2025; Favero et al., 2025c). This perspective views memorization primarily as a global phenomenon with samples being memorized approximately synchronously once some threshold is crossed.

Disentangling these competing hypotheses is further complicated by the absence of a unified definition and standardized evaluation of memorization (Schwarzschild et al., 2024). Often, due to computational constraints, matching generated data to elements in the train set involves finding memorization candidates (Carlini et al., 2023a) or performing evalua-

tions on smaller subsets of the train set (Aerni et al., 2025). Moreover, the boundary between memorization and generalization is not always clearly defined, as researchers use metrics that are likely sensitive to properties of the underlying data. For instance, definitions that label a text output as memorized if it reproduces an entire training example or if it matches some fixed minimum number of characters verbatim (Carlini et al., 2023b; Aerni et al., 2025; Nasr et al., 2025) are highly dependent on the text's length and rarity.

Furthermore, the analysis of memorization is complicated by the fact that natural data (images, text) is compositional in nature. Images, for instance, are comprised of objects, and those objects of parts, and so forth. This means that a generative model exhibiting *partial memorization* could engage in collage, by composing *memorized fine-grained features* into novel arrangement. However, because reliably identifying and matching such sub-features in natural data is difficult, most existing memorization studies primarily focus on reproduction of entire training examples (Carlini et al., 2021; Tirumala et al., 2022; Carlini et al., 2023a) with only a small number of recent works referring to concepts tangential to partial memorization (Favero et al., 2025c; Di et al., 2026).

Here, we address this challenge of distinguishing generalization, partial memorization, and full memorization, by studying the memorization process of diffusion models (Sohl-Dickstein et al., 2015; Ho et al., 2020) trained on explicitly hierarchical and compositional synthetic data generated by the Random Hierarchy Model (RHM) (Cagnetta

et al., 2024). To address our question of how *sample rarity* enters into memorization, we consider a recent extension of the RHM (Cagnetta et al., 2025) that couples the data generation rules to a power-law probability distribution. By controlling the level of abstraction at which this power-level enters, we can probe for the first time the interaction of rarity and abstraction on memorization and partial memorization.

Our findings can be summarized as follows:

- We observe *preferential memorization* even in deduplicated scenarios, with certain training examples at risk of earlier memorization than others. In particular, examples that can be expressed as combinations of common sub-features are more likely to be generated early.

- We provide a simple argument linking the propensity to memorize outliers to the difficulty of memorizing fat-tailed distributions. Consequently we find that heavy-tailed datasets delay the process of memorization (cf. Figure 1-*bottom*), particularly when the heavy-tail is introduced at high-levels of abstraction.

- We show that during the initial stages of memorization, common features are over-represented amongst the generated data. Thus, there exists a regime of *partial memorization*. Halting training in this regime would result in models being biased towards generating these common features, i.e. outputting "slop".

- We verify that the phenomena of *preferential memorization* and *partial memorization* are general; they also occur in diffusion models trained on image data.

## 2. Related work

**Distributional properties of generative models** Recent work shows that language models overproduce syntactic templates (relatively abstract compositions of text) in comparison to human-generated text (Shaib et al., 2024). The paucity of output diversity, a form of distributional shift, is credited with model collapse (Dohmatob et al., 2024) and can be viewed as a generalization-to-memorization transition (Shi et al., 2025).

**Properties of memorized data** Previous work has shown that data duplication is a major cause of memorization in transformer-based language models, both under adversarial (Lee et al., 2022; Carlini et al., 2023b; Prashanth et al., 2025) and non-adversarial prompting (Aerni et al., 2025). In the context of diffusion-based image generation, Carlini et al. similarly show that duplicate images are more easily extracted from trained models. However, they observe that duplication alone does not fully account for the observed memorization patterns. Beyond duplication, several studies find that rare or atypical texts are overrepresented among

memorized samples (Carlini et al., 2023b; Aerni et al., 2025; Morris et al., 2025). Additionally, Speicher et al. show that higher-entropy training strings enter a memorization phase earlier than lower-entropy strings, suggesting that entropy accelerates the transition from generalization to memorization. Our work sheds light on memorization beyond deduplication, and provides a fine-grained characterization of memorized data points by analyzing how sub-components of an example make it more likely to be preferentially reproduced.

**Memorization metrics** Existing literature employs a range of notions of memorization, reflecting different definitions and measurement choices (Schwarzschild et al., 2024). In text, it is common for works to primarily evaluate memorization through verbatim reproduction, (Lee et al., 2022; Carlini et al., 2023b; Prashanth et al., 2025; Aerni et al., 2025). In contrast, Morris et al. adopt an information-theoretic definition of memorization by quantifying the amount of information a model retains about a data point after training. Their approach additionally distinguishes between intended and unintended memorization, where the former corresponds to overlap arising from legitimate generalization. Moreover, most existing metrics for memorization in diffusion generation focus on whole samples, and therefore can miss subtler memorization patterns. To address this, recent work proposes segmentation-based metrics that quantify memorization at the level of foreground and background regions, enabling finer-grained analysis of how diffusion models reproduce specific parts of training images (Di et al., 2026). Our work expands this notion within a synthetic framework, enabling analysis of memorization at the level of smaller components of data points, while also performing exhaustive evaluations over the training dataset and, where necessary, accounts for the expected overlap observed in data under *unbiased* generation.

**Memorization dynamics** In diffusion models, it is known that the optimal empirical score is attained by complete memorization of the training data (Gu et al., 2025b; Baptista et al., 2025). Additionally, studies have observed that training in diffusion models exhibits three phases: (i) an initial phase of generalization, followed by (ii) an increase in validation loss as score starts to converge to the empirical score, before finally (iii) a memorizing phase, in which samples are exclusively generated. The transition to memorization occurs at earlier training steps for larger models and smaller datasets (Pham et al., 2026; Gu et al., 2025a; Bonnaire et al., 2025; Favero et al., 2025c). These results suggest that early stopping when validation loss starts to increase can mitigate memorization by preventing the transition into the fully memorizing regime. However, existing analyses primarily operate at the level of entire data points

and ignore the presence of rare features and samples. In contrast, our approach examines the evolution of partial memorization and reveals that halting training even before total memorization can result in a systematic bias towards reproducing the most common sub-features in the training data, i.e. slop.

# 3. Preliminaries

## 3.1. Diffusion models

Diffusion models (Sohl-Dickstein et al., 2015; Ho et al., 2020; Song et al., 2021b) model a data distribution from a collection of samples by learning to gradually denoise data. During training, a forward noising process is applied to a data sample $\mathbf{x}_0$ via a Markov chain $q(\mathbf{x}_{1:T} \mid \mathbf{x}_0) = \prod_{t=1}^{T} q(\mathbf{x}_t \mid \mathbf{x}_{t-1})$, which progressively adds noise such that at the final timestep $T$, $\mathbf{x}_T \sim p_T \approx \mathcal{N}(\mathbf{0}, \mathbf{I})$. The generative model is defined by a reverse process that learns transitions $p_\theta(\mathbf{x}_{t-1} \mid \mathbf{x}_t)$, enabling generation by iteratively denoising from noise. In the score-based formulation, this reverse process is parameterized by a neural network $s_\theta(\mathbf{x}_t, t)$ trained to approximate the score $\nabla_{\mathbf{x}_t} \log q(\mathbf{x}_t)$ of the noised data distribution via a denoising score-matching objective (Song et al., 2021b).

Several extensions of diffusion models have been proposed for discrete settings (Hoogeboom et al., 2021; Austin et al., 2021). In our work, we implement multinomial discrete diffusion models, in which the forward process is a Markov chain that progressively replaces tokens in the sequence with a randomly sampled symbol. The reverse process is then trained to iteratively reconstruct the original sequence by denoising from these partially or fully corrupted sequences. We use a U-Net architecture (Ronneberger et al., 2015) with weight sharing, with code based on (Favero et al., 2025b). See Appendix A.1 for full model details.

## 3.2. Random Hierarchy Model

Natural data is inherently compositional. In vision, this idea is often formalized as Pattern Theory (Siskind et al., 2007; Grenander, 1996; Jin & Geman, 2006), which models a scene as a hierarchy: scenes consist of objects, objects consist of parts, and parts decompose into sub-parts, continuing down to basic visual primitives such as facets, edges, and colours (cf. Figure 2). Language exhibits a parallel structure. Syntactic parse trees represent sentences hierarchically: sentences are composed of phrases, phrases of sub-phrases and so forth (Chomsky, 1965).

The Random Hierarchy Model (RHM) is a synthetic model for data that exhibits this hierarchical compositionality (Cagnetta et al., 2024). A realization of the RHM is formally a probabilistic context-free grammar defined by a set of fixed production rules for decomposing symbols into tuples of sub-symbols, and sub-symbols into tuples of sub-symbols (see Figure 2: *Right*). The production rules are randomly selected at initialization and fixed throughout data generation. The task of diffusion models trained on such data is to generate new data consistent with these fixed rules. For every symbol $v_i^{(\ell)}$ at depth $\ell < L$ of the tree ($\ell = 0$ corresponding to the root, and $\ell = L$ to the leaves) and position $i \in (1, s^\ell)$, there are $m$ distinct production rules (synonyms) to legally decompose the symbol into a tuple of $s$ sub-symbols.

The objective of this paper is to characterize how sample likelihood drives memorization, and how this effect interacts with the compositional structure of natural data. In the baseline RHM however, as all production rules are equiprobable, all valid data are equally likely: the model does not encode any preference for some samples over others and there is no notion of sample rarity. We therefore consider an extension to the RHM first introduced in (Cagnetta et al., 2025), where production rules have non-uniform sampling frequencies. Here, we use Zipfian sampling frequencies only at level $\ell = \ell_z$, selecting production rule $m' \in \{1, 2, \ldots m\}$ with probability $p(m') \sim m'^{-\alpha}$ for fixed exponent $\alpha$, making the generative process preferentially sample earlier rules. For all other tree-depths, $p(m'; \ell \neq \ell_z) = 1/m$ and production rules are sampled uniformly, as in the standard RHM.

Both $\alpha$ and $\ell_z$ are control parameters we will vary in Section 4.2. Unless otherwise noted, we fix the RHM parameters to $L = 4$, $v = 6$, $m = 4$, $s = 2$, and train models with $N = 2 \times 10^4$ samples, which is sufficient to achieve generalization, but much smaller than the maximum number of legal data $N_{\max} \approx 6.44 \times 10^9$ which makes identification of memorization easy. We choose these parameters to minimize computational requirements while enabling the study of partial memorization. See Appendix A.3 for extended details.

## 3.3. Partial memorization

In the literature, one of the most common ways to quantify memorization is by evaluating whether generated samples appear verbatim in the training data (Lee et al., 2022; Carlini et al., 2023b; Prashanth et al., 2025; Aerni et al., 2025). In line with this, we use the fraction of copies generated as a first indicator of memorization.

**Definition 3.1** (Complete memorization). Given $N_t$ sampled data points, of which $N_c$ are exact copies of elements in the train set, the fraction of copies is given by $\frac{N_c}{N_t}$.

Memorization may also occur at a finer granularity, with models preferentially reproducing components or substrings of training data prior to full-sequence copying. In this regime, exact copies may be rare or absent, yet the model's sampling behavior may still be biased towards certain training data points. We study this phenomenon by considering

**Hierarchical Composition in Images:**

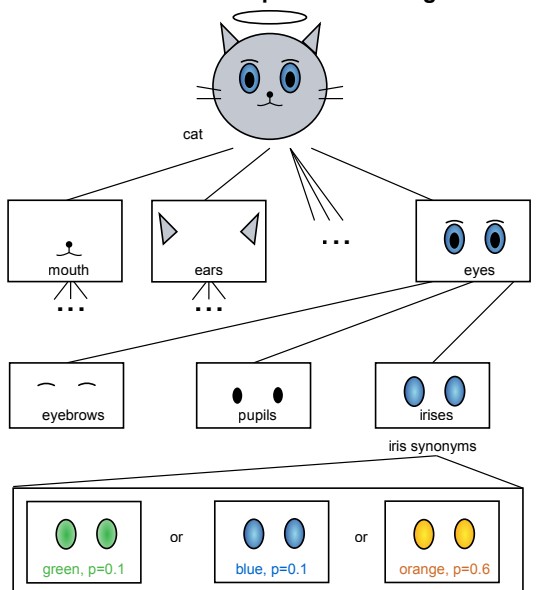

**Example RHM sample**

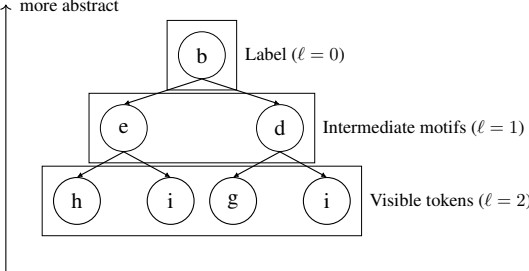

more abstract

Label ($\ell = 0$)

Intermediate motifs ($\ell = 1$)

Visible tokens ($\ell = 2$)

less abstract

**Example RHM rules**

depth: L=2, vocabulary-per-level: v=3, synonyms: m=3, branching ratio: s=2

| Levels | Production Rules | | |
|---|---|---|---|
| $\ell = 0 \to \ell = 1$ (unequal rule probabilities) | a — df 0.6, dd 0.3, fd 0.1 | b — ff 0.6, ee 0.3, ed 0.1 | c — de 0.6, ef 0.3, fe 0.1 |
| $\ell = 1 \to \ell = 2$ (equal rule probabilities) | d — hh, gi, hg | e — gg, hi, ih | f — ig, ii, gh |

*Figure 2.* **The random hierarchy model captures the hierarchical compositionality of images.** *Left*: Images are comprised of parts, and those parts of simpler parts, and so forth, engendering images with a natural hierarchy. Additionally, many motifs exhibit equivalent variants, as exemplified here by the different colours of irises that could be sampled by a putative generative process. *Right*: The RHM is an idealization of this process. A part $x^{(\ell)} \in \{1, 2, \ldots, v\}$ at level $\ell$ of the hierarchy is comprised of $s$ sub-parts $(x_1^{(\ell+1)}, x_2^{(\ell+1)}, \ldots, x_s^{(\ell+1)})$ at level $\ell + 1$. There are $m$ valid sub-tuples (*synonyms*) with which to decompose a given part $x^\ell$.

the generation of valid subtuples under a given rule. However, even a model that samples fairly according to the underlying RHM rules may generate tuples that appear in the training set, since this constitutes a subset of all valid tuples. When this subset covers a significant fraction of all possible valid tuples, observing overlap with the training set alone is insufficient to conclude memorization.

To disentangle fair sampling from memorization, we model tuple generation as a mixture process. Let $v'$ denote a symbol at level $\ell \leq \ell_z$, and let $m'$ denote an associated rule. We assume that, with probability $\lambda \in [0, 1]$, the model copies a tuple from the training set, and with probability $1 - \lambda$ it samples a tuple uniformly from the space of valid tuples defined by the RHM. Since $\ell \leq \ell_z$ with fixed $m'$, all such tuples are equiprobable. Let $P_{\text{copy}}$ denote the probability that a copied tuple appears in the training set (equal to 1 by construction), and let $P_{\text{fair}}$ denote the probability that a tuple sampled uniformly at random appears in the training set. Under this model, the probability that a generated tuple appears in the training set is

$$P\left(\{v_i^{(L)}\}_{i=1}^{s^{L-\ell}}, v', m'\right) = \lambda P_{\text{copy}} + (1 - \lambda) P_{\text{fair}}.$$

The probability $P_{\text{fair}}$ depends on the coverage of the training set over the space of valid tuples for the rule $(v', m')$. We define

$$f(v', m') = \frac{\text{\# unique tuples in train set for rule } (v', m')}{\text{\# total valid tuples for rule } (v', m')},$$

so that $P_{\text{fair}} = f(v', m')$.

Given a model that generates $N_g$ valid tuples, of which $N_c$ appear in the training set, the expected number of such tuples is $N_c = N_g(\lambda + (1 - \lambda)f(v', m'))$. Rearranging yields an estimator for the memorization parameter $\lambda$.

**Definition 3.2** (Partial memorization). The estimator

$$\lambda = \frac{N_c - N_g f(v', m')}{N_g(1 - f(v', m'))}$$

quantifies the extent to which the model's behavior deviates from fair sampling towards memorization.

Values of $\lambda > 0$ indicate partial memorization even in the absence of complete memorization. More broadly, for a general sampler that outputs training samples with probability $P_{sampler}$, $\lambda$ can be rewritten as

$$\lambda = \frac{P_{\text{sampler}} - P_{\text{fair}}}{1 - P_{\text{fair}}}.$$

Thus, $\lambda > 0$ detects any excess reproduction of training samples beyond that explained by the ground-truth generative process, whether arising from explicit memorization or other distributional biases. In this sense, the estimator is conservative: anomalous overproduction of training samples is always ascribed to copying.

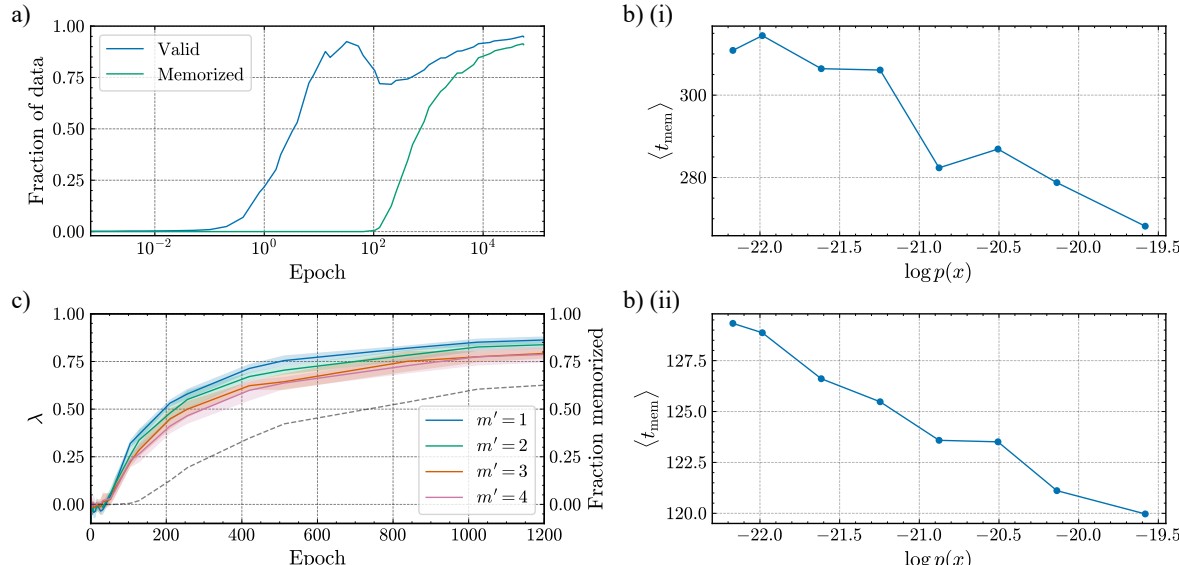

*Figure 3.* **Rare data are memorized later.** Evolution of memorization for models trained with $\ell_z = 1$. **a)** Fraction of valid and completely memorized samples as training progresses. **b)** First time $t_{\mathrm{mem}}$ a train sample is generated when sampling $10^5$ data points per checkpoint, averaged across elements of similar log-likelihoods for **(i)** width 1024 and **(ii)** width 4096 models. **c)** Evolution of partial memorization per rule for the width 1024 model, averaged across rules of the same frequency. The dashed line shows the fraction of output samples in the training set, as in panel **a**.

## 4. Results

We now train our diffusion models (see Appendix A.1 for training details) in a variety of settings to first test the effect of rarity on the memorizability of individual samples (Section 4.1) and then on the memorizability of datasets (Section 4.2). Finally, we study the dynamics of early phases of memorization (Section 4.3).

### 4.1. Common data are memorized first

We begin with analyzing the relation between the time to memorization and the rarity of samples in the train data. We define the *time to memorization* $t_{\mathrm{mem}}$ of a sample as the earliest training step at which that sample appears at least once among the generated output. To characterize the *rarity of samples*, we use their log-likelihood, calculated as the sum of the log-probabilities of the rules sampled to generate each data point.

Figure 3a shows a representative example of the training dynamics observed across models: initially, models learn the implicit rules of the RHM, successfully generating valid samples that are not found in the training set. At later stages of training, the model begins to reproduce elements of the training data, leading to an increase in memorized samples. This behavior is consistent with known results in the literature (Pham et al., 2026; Bonnaire et al., 2025; Favero et al., 2025c). In particular, we observe that the time window for generalization preceding memorization is reduced as model size increases (below a certain capacity threshold) in Appendix B.3.

**On average, common training elements are generated earlier during training.** Figure 3b shows a negative linear relationship between the log-likelihood of data points and $\langle t_{\mathrm{mem}} \rangle$, time to memorization, averaged across data points of similar log-likelihood values. Since the training dataset consists of only unique samples, this relationship cannot be explained by differences in frequency and instead reveals that common data points are more susceptible to memorization, as predicted by **Hypothesis 2**. These results are consistent across different model sizes, as well as for Zipf introduced at layers of the hierarchy below root level ($\ell_z = 1, 2, 3$).

**Memorization proceeds in stages, from lower-level features to full data points.** Figure 3c shows that memorization proceeds in distinct stages: during the early phases of training, there is an increase in the partial memorization of all $s^{L-1}$ subtuples, even before any detectable memorization of full data points (marked with a dashed line in Figure 3c) occurs. In practice, this implies that the absence of exact sample reproduction does not guarantee the absence of memorization, and that partial memorization provides an earlier indicator of memorization risk.

**Frequent rule subtuples are memorized faster than rare rule subtuples.** In addition to the emergence of partial memorization, Figure 3c reveals a tendency of models towards memorizing subtuples from the most frequent rules. As a result, models do not only preferentially memorize common data points, but also their underlying common features.

This will reflect in a bias of models towards overproducing a limited set of concepts, limiting the feature-level diversity of samples generated, even when these are distinct. This effect is exaggerated for smaller model sizes, as we show in Figure 13.

### 4.2. Memorization and data distribution

We have established that rare samples are harder to memorize, i.e. are memorized later. Next, we turn our attention to the question of which data distributions are harder to memorize, and connect this to our understanding of sample memorization. An understanding of which datasets are easier to memorize is useful, because it is (i) at the dataset level that that training decisions are typically made, and (ii) easier to quantify variability / presence of tails at the dataset level than it is to quantify rarity at the sample level in natural data.

**Fat-tailed distributions are harder to memorize.** To measure the difficulty of memorizing a dataset, we consider the mean sample memorization time, $\langle t_{\mathrm{mem}} \rangle_D$. We treat the full analytic calculation for this quantity in Appendix C.1, but for brevity's sake consider here a simplified setting in which the time to memorization for a sample of rarity $r$ scales as $t_{\mathrm{mem}}(r) \sim p(r)^{\beta}$ where $\beta < 0$ implies that common samples are memorized first (as implied by Figure 3), while $\beta > 0$ implies that rare samples are memorized first. If the data are exponentially distributed, with $p(r) \sim e^{-\alpha r}$, where $\alpha$ characterizes the rarity scale (with larger $\alpha$ implying fewer rare samples), then straightforward integration yields $\langle t_{\mathrm{mem}} \rangle_D \sim \frac{\alpha^{\beta}}{1+\beta}$. This implies that for $\beta < 0$, $\langle t_{\mathrm{mem}} \rangle_D$ should decrease with increasing $\alpha$, which we confirm in Figure 4. Note that the pattern observed (inset in Figure 4) matches the predicted outcome of Figure 1-*bottom* for the case where common features are memorized first.

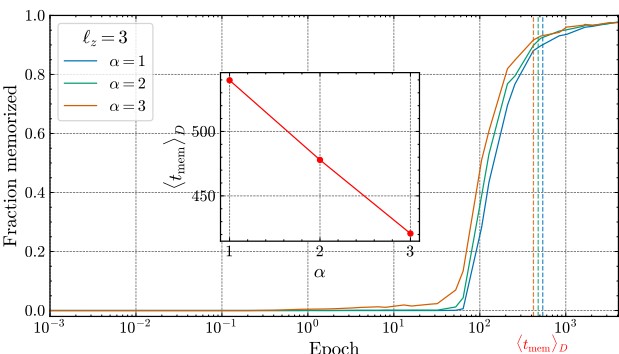

*Figure 4.* **Broad distributions are memorized more slowly**. Evolution of fraction of memorized samples for models of width 4096 trained with Zipf distributions of varying exponent $\alpha$. *Inset:* time to memorization $t_{\mathrm{mem}}$, averaged across all memorized elements of the training set.

**Class variation is harder to memorize than leaf variation.** It's well known that balancing class distributions improves the performance of trained classifiers. How important is this at different levels of abstraction? Do rare abstract features (such as class variation) or rare concrete features (such as unusual textures) have a greater impact on the memorizability of datasets? We test this in Figure 5, by introducing rarity to the sampling rules at different depths of the tree $\ell_z$. Models trained with different $\ell_z$ generalize at approximately the same time (cf. Figure 5 top) but those trained with more abstract variation (i.e. smaller $\ell_z$) memorize later. We conjecture that this is in part because the model sees proportionally more ($s^{L-1}$ more) examples of rare features when introducing Zipf's law at $\ell_z = L - 1$ than when it is introduced at the class level of $\ell_z = 0$.

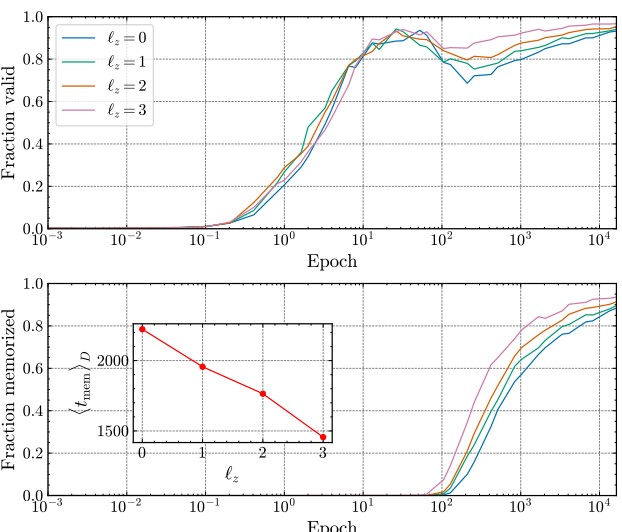

*Figure 5.* **Diffusion models memorize class variation slower than leaf variation**. Training evolution for models of width 1024 trained with Zipf distribution with exponent $\alpha = 2$. *Top:* Models trained with differing $\ell_z$ generalize at a similar epoch. *Bottom:* Models with rare classes $\ell_z = 0$ exhibit delayed memorization. Inset: time to memorization $t_{\mathrm{mem}}$, averaged across all memorized elements of the training set.

### 4.3. Memorization dynamics

Figure 5 shows a temporary decrease in valid samples across all models as memorization begins. We hypothesize that this decrease in valid samples reflects a *repurposing of model capacity*. Under this assumption, a model with larger width (and parameter count) will have excess capacity with which to memorize more data points without harming the generation of valid samples. We observe this behavior in Figure 10), where the drop in performance is ameliorated in larger models. As memorization begins, neurons previously supporting generalizable structure are reallocated to encode training examples, temporarily impairing the model's ability to generate valid outputs until memorization (by definition counted as valid) restores the valid sample count.

To better understand this phenomenon, we analyze the model's sampling distribution throughout training. For a given symbol $v' \in \{1, 2, \ldots, v\}$ from level $\ell_z$, let $P_{v'}$ denote the empirical distribution over rules in the model's generated samples and $Q_{v'}$ the corresponding distribution of rules in the training set. We compute the average discrepancy across symbols as $\frac{1}{v} \sum_{v'=1}^{v} D_{KL}(P_{v'} \| Q_{v'})$.

Results for models of width 4096 are presented in Figure 6. Similar results are observed across different model sizes (see Figure 14).

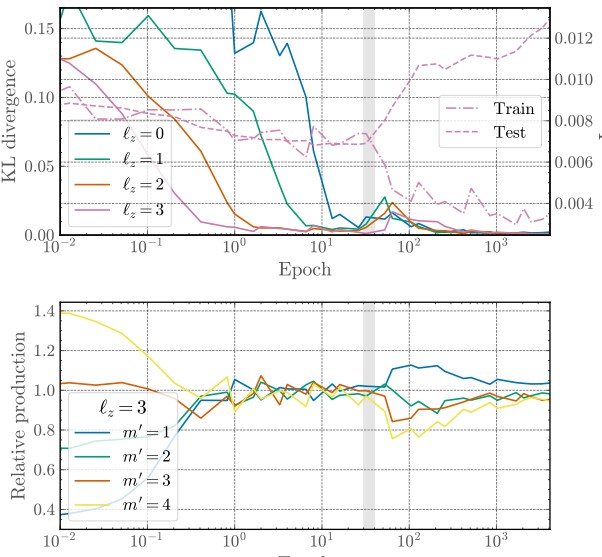

*Figure 6.* **Memorization transition induces distributional shifts.** *Top:* evolution of the average KL divergence between the empirical distribution of generated subtuple rules corresponding to $\ell = \ell_z$ for models trained with different $\ell_z$. The dashed curves indicate the train and test losses as a function of training epochs for the case $\ell_z = 3$. *Bottom:* ratio of actual production rule usage to the expected rate, as a function of training epoch for $\ell_z = 3$ (similar findings hold for other $\ell_z$). Excess production of the common rule and underproduction of rarer rules precedes memorization.

**The initial stage of memorization coincides with a degradation of the learnt distribution.** Figure 6 (top) shows an initial decrease in the KL divergence between the empirical generated and training distributions across all models, reflecting a stage of generalization in which the models learn to generate the Zipf-distributed rules. As training continues, this trend reverses: the KL divergence increases precisely when the fraction of valid samples decreases and the fraction of memorized samples starts to grow. This behavior reflects a breakdown in distributional learning: the model deviates from the target distribution, resulting in a decrease in generative performance. This effect is most pronounced for models trained with Zipf's law introduced at levels $\ell_z = 1, 2, 3$.

**Common rules are overproduced at the start of memorization.** For these levels, comparing the number of sub-

tuples generated for each rule frequency at level $\ell_z$ to the rule frequency the model is expected to generate when accurately learning the distribution, reveals an increase in the most frequent subtuples which coincides with a decrease in the generation of rarer rule subtuples. We dub such overproduction *"slop"*. We show this behavior for $\ell_z = 3$ in Figure 6. Analogous results for the remaining Zipf levels are in Appendix B.4. This pattern suggests that the neurons supporting rarer rules are being preferentially repurposed during this stage of training.

**Early stopping can avoid overproducing common subfeatures.** Figure 6 (top) shows the evolution of train and test losses of the $\ell_z = 3$ model. Early stopping is commonly applied at the stage of training where train and test losses diverge (Favero et al., 2025c; Bonnaire et al., 2025). Our results show that the increase in the test loss generally precedes the increase in KL divergence, thus avoiding the bias of common features in the transition to memorization. Despite this, we highlight the importance of careful checkpoint selection, as variations in the stopping range can easily place the model under partial memorization and overproduction of common features (see Figure 15).

## 5. Experiments on real data

In this section, we validate our observations on real-world datasets by training diffusion models on subsets of CelebA (Liu et al., 2015). We take checkpoints throughout training, and at each checkpoint sample 50,000 output images (see Figure 18 for representative outputs). To monitor performance we use the FID score (Heusel et al., 2017) against a test set of 50,000 images from CelebA. We identify memorized images using the criterion of (Bonnaire et al., 2025), and report the fraction of memorized images generated as a measure of memorization.

Further, each image in CelebA has assigned True/False labels corresponding to 40 attributes (e.g., "bald", "wearing lipstick"). By fitting a likelihood estimator to the statistics of these attributes, we can assign likelihoods to training images. For samples generated by the model that do not have assigned labels, we train a multi-label classifier to estimate the values of these, and infer a corresponding likelihood. Further details can be found in Appendix D.

**Images composed of common features are memorized earlier.** We train a model on a subset of 10,000 images and evaluate the relationship between time to memorization and log-likelihood. Figure 7 shows that "common" (higher log-likelihood) training images are reproduced at earlier stages in training. Additionally, we confirm this behavior on a smaller 1,000 image subset where the model reaches 98% memorization in Figure 16.

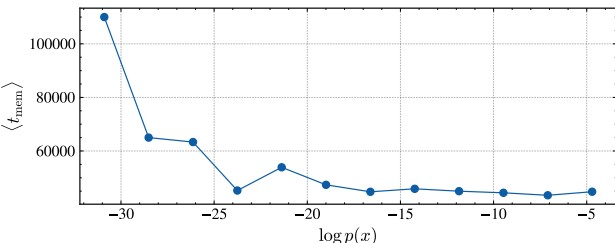

*Figure 7.* **Rare images are memorized later.** Training time to memorization $t_{\mathrm{mem}}$ (in epochs), averaged across train images of similar estimated log-likelihood, for a model trained on a 10,000 image dataset achieving 29.2% memorization. See Appendix D.2 for experiment details and Figure 8 for training dynamics.

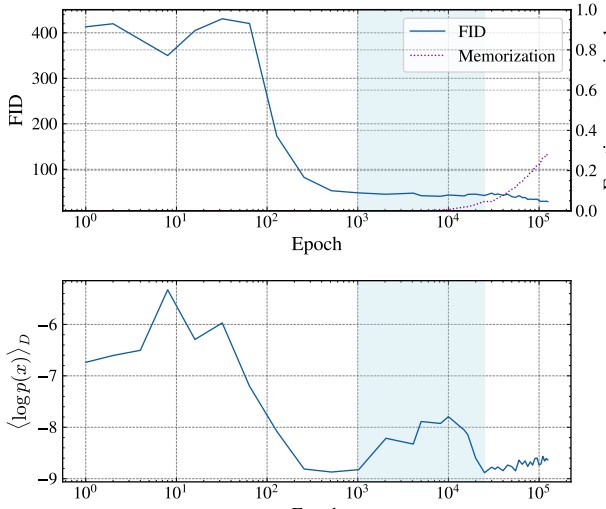

*Figure 8.* **There is a stage of overproduction of common features before memorization starts.** *Top:* evolution of the FID score (generalization) and the fraction of images memorized throughout training, for a model trained on a dataset of 10,000 images. *Bottom:* average estimated log-likelihood of generated images throughout training. The grey region denotes the pre-generalization phase, as assessed by FID; the blue region denotes the increase in log-likelihood "slop" phase. See Appendix D.2 for experiment details.

**Memorization is preceded by a "slop" generating phase.** Figure 8 shows that after generalization (as indicated by FID), but before full memorization, the model produces outputs composed of more common features, just as in the RHM (cf. Figure 6). This again meets the distribution shift definition of "slop".

**Smaller models exhibit a longer "slop" phase.** In Figure 17 we compare the training dynamics for two models of different sizes (63M vs. 109M parameters). Memorization begins later and proceeds slower for the smaller model, while the "slop" phase is prolonged. This agrees with observations in the RHM (cf. Appendix B.4). Intriguingly, the "slop" phase in the smaller model exhibits increased FID – FID may help identify a "slop" phase for smaller models but fail for larger ones.

## 6. Discussion

Although diffusion models are susceptible to memorization given sufficient training, it does not occur uniformly across samples. We show for both synthetic and natural data that *preferential memorization* progresses in distinct stages. The model begins reproducing common low-level features, followed by more abstract, higher-level common features. As training continues, samples comprised of more common features are memorized earlier than those composed primarily of rarer features, thereby resolving the ambiguity of whether common or outlier data points are memorized first in favor of Hypothesis 2 (cf. Figures 1 and 3).

We identify characteristics that determine whether a dataset is more at risk of memorization: more diverse feature distributions are less susceptible to memorization than more concentrated distributions. In addition, variability and biases at finer feature-level representations have a stronger impact on memorization than variability at the level of full data points. This highlights the importance of dataset curation and ensuring a sufficiently diverse distribution at different levels of abstraction.

We also identify for synthetic and natural data a *partial memorization* regime in which the model overproduces common training-set features, a distributional shift denigrated as "slop". Although this regime often emerges when train and test losses diverge, the two can occur close in time. As a result, infrequent checkpointing may overshoot this boundary, placing models in the partial memorization regime despite early stopping. The staged nature of memorization also implies that data point–level metrics will fail to detect this partial memorization, where models reproduce substantial subsets of a sample's features without memorizing the full sample. This motivates the need for finer-grained evaluation metrics. We note that lower capacity models, preferred as a means to reduce memorization, can amplify these biases.

**Limitations and future work** Compared to feature-level variation, class-level variation delays memorization onset (Section 4.2), while reducing preferential memorization (cf. Figures 11 and 12 and associated discussion). Disentangling these effects experimentally and theoretically would be a valuable future contribution. Moreover, our $\lambda$ metric for partial memorization and the sample likelihood estimator require an explicit labeling of the latent variables (attributes) underpinning the data, limiting applicability when such labeling is intractable. Another open question is whether the observed bias toward common samples and the sequential nature of partial memorization also appear in next-token prediction models. More broadly, it remains unclear whether partial memorization can emerge in training regimes where higher-level abstract generalization is still improving (e.g., the model combines coarse-grained features in novel ways, while relying on memorized copies of those same features).

## Acknowledgements

We thank Alessandro Favero for fruitful discussions and for feedback on an early draft of this manuscript. We thank Alessandro Favero and Antonio Sclocchi for the initial RHM diffusion model code. D. J. Korchinski acknowledges financial support from the Natural Sciences and Engineering Research Council of Canada (NSERC PDF - 587940 - 2024). M. Aparicio Rodriguez acknowledges financial support from the Department of Mathematics at Imperial College London through the Roth Scholarship and from the ML Lab at Capital Fund Management.

## Impact Statement

This paper presents work whose goal is to advance the field of Machine Learning. There are many potential societal consequences of our work, none which we feel must be specifically highlighted here.

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

# A. Methods

## A.1. Architecture

We use a 1D U-Net architecture with four downsampling blocks and four upsampling blocks, i.e. matched to the $L$ of the RHM we study, as was introduced in (Sclocchi et al., 2025a;b). All convolutional layers operate on a fixed number of feature channels (width), which remains constant at every resolution level. In each layer, we implement weight sharing across spatial positions. We evaluate models using widths of 1024 and 4096 channels. The total number of learnable parameters for each is 25M and 409M, respectively. We train all models using Adam and a learning rate of 0.01. Code is available at https://github.com/martaaparod/memorization-in-diffusion-models.

## A.2. Sampling

Unless stated otherwise, each model checkpoint is evaluated for memorization and partial memorization by generating $10^4$ samples and checking which fraction is in the training set.

## A.3. Training set

We train our diffusion models on a fixed set of RHM rules with parameters $L = 4$, $v = 6$, $m = 4$, $s = 2$, as discussed in Section 3.2. The training set consists of 20,000 unique samples generated by the RHM. When $\alpha$ or $\ell_z$ are changed, these samples are re-drawn according to the updated production rule probabilities. The resulting distribution of log-likelihoods is visualized in Figure 9.

We chose the RHM parameters with two objectives: (i) to minimize computational expense, and (ii) choose an RHM grammar with sufficient richness such that even when the training set size $N$ exceeds the sample complexity of generalization $N^*$, there are still un-sampled substrings of length $s^{L-1}$, so that we can detect partial memorization.

The training set size $N$ is bounded below by the sample complexity $N^* \sim vm^L$ (Favero et al., 2025a). The separation between the training set size $N$ and $N_{\max}$ is crucial, so that we can distinguish generalization (when novel samples are generated) from full and partial (sub-tuple) memorization (discussed in Section 3.3). Similarly, we also want to ensure that the training set does not fully cover the $s^{L-1}$ sub-tuples. For our choice of RHM parameters, there are $N^{(L-1)} = vm^{(s^{L-1}-1)/(s-1)} = 98,304$ unique sub-samples, of which the majority are unobserved in the training set. The number of legal sub-tuples of size $s^\ell$ grows asymptotically as $m^{(s^{L-\ell}-1)/(s-1)}$, which is much faster than the sample complexity $m^L$. Were greater computational resources available, we could select $s = 3$ or $L = 5$, and our partial memorization results could be extended to probe still smaller tuples of scale $s^{L-2}$ or even $s^{L-3}$.

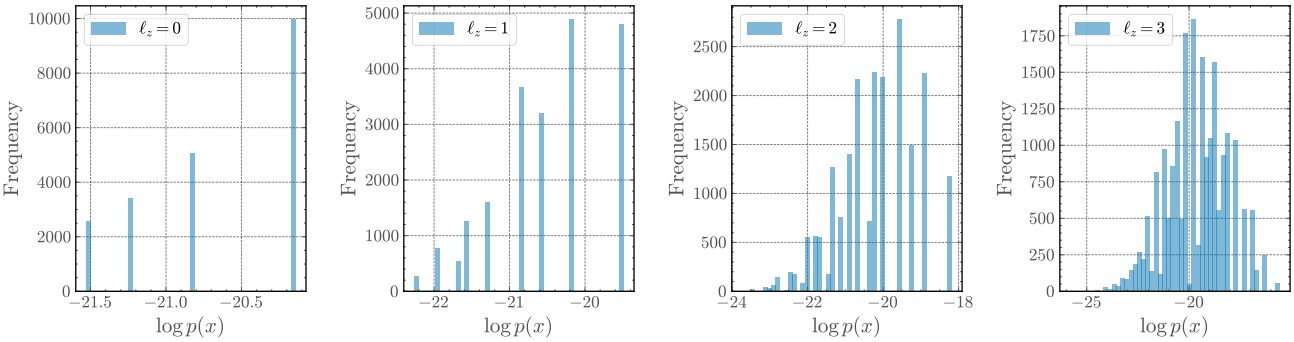

*Figure 9.* **Log-likelihood frequencies are lognormal for $\ell_z \gg 0$.** Histograms represent linearly binned log-likelihoods for $N = 2 \times 10^4$ samples drawn with Zipf-exponent $\alpha = 1$.

.

# B. Additional results

In this section, we provide additional graphs to showcase model behavior under varying degrees of model capacity. We present the results for the widths 1024 and 4096 models, and additionally present results for underparametrized models (width 256).

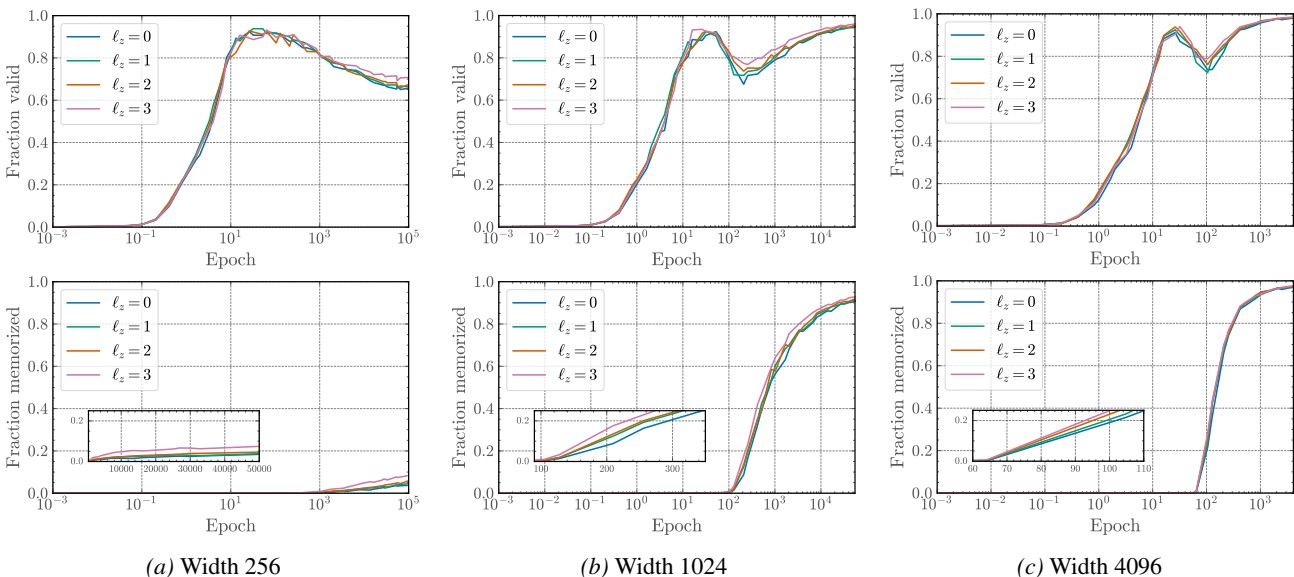

*Figure 10.* **Models first generalize and then memorize.** Evolution of training dynamics for $\alpha = 1$ and models with widths **a)** 256, **b)** 1024, and **c)** 4096. *Top:* fraction of valid samples. *Bottom:* fraction of memorized samples.

### B.1. Training dynamics

**Generalization precedes memorization for a shrinking window as model capacity increases.** Figure 10 shows that for all models, the valid count of samples increases before memorization increases. In particular, this increase is slightly delayed for larger-sized models. Conversely, the memorization curve increases significantly earlier, and at a faster rate for models with higher width, with the 4096 model achieving more than $95\%$ data point-level memorization in 4000 epochs, while the 256 model barely achieves $10\%$ memorization after 100000 epochs. These observations align with existing results in the literature with regards to the generalization window of diffusion models (Pham et al., 2026; Bonnaire et al., 2025; Favero et al., 2025c).

**The differences between the memorization curves of class and leaf variation diminish as model capacity increases.** We observe that for a lower value of the Zipf exponent $\alpha$, the differences between the memorization curves for different $\ell_z$ is less noticeable than in Figure 5. Additionally, as model width increases, we can see that the range of epochs over which these differences are observable progressively narrows. Practically, this suggests that dataset properties remain an important factor in the training dynamics of models, most importantly in the case of underparametrized models.

### B.2. Time to sample memorization versus likelihood

We present in Figure 11 the relation between time to memorization and the log-likelihood of elements in the train data for models with width 1024 and 4096. Complementing this, we study in Figure 12 the average likelihood of regurgitated training examples, and find a gradual shift towards rarer samples at late times.

**Memorization of common features is independent of rarity at root level.** Figure 11 shows that at level $\ell_z = 1, 2, 3$, the data points composed of common features are memorized earlier than those composed of rare features. We note that as data points become more rare (smaller likelihood), the value of $\langle t_{\mathrm{mem}} \rangle$ becomes more irregular due to a smaller number of samples for those log-likelihood values. Interestingly, the memorization of common data points is not preferentially memorized for the case $\ell_z = 0$, suggesting that higher-level biases do not impact as strongly the memorization of data points. It is worth noting that at higher levels of abstraction the range of $t_{\mathrm{mem}}$ values across which we observe this effect gradually decreases, and thus our chosen hyperparameters, such as checkpoint frequency or model size, may be insufficient to detect it. Conversely, this could reflect the fact that diffusion models learn the latent hierarchy at level $\ell$ from the correlations between the subordinate tuple below a latent at level $\ell + 1$, with tokens outside the descendants of that tuple (Sclocchi et al., 2025a). Concretely, the root node is singular in the hierarchy, there is no correlation signal for the diffusion model to exploit at this level – diffusion models do not construct a latent corresponding to the class (Korchinski et al., 2026). This may explain the distinct behavior observed at the root. We leave analysis of this behavior to future work.

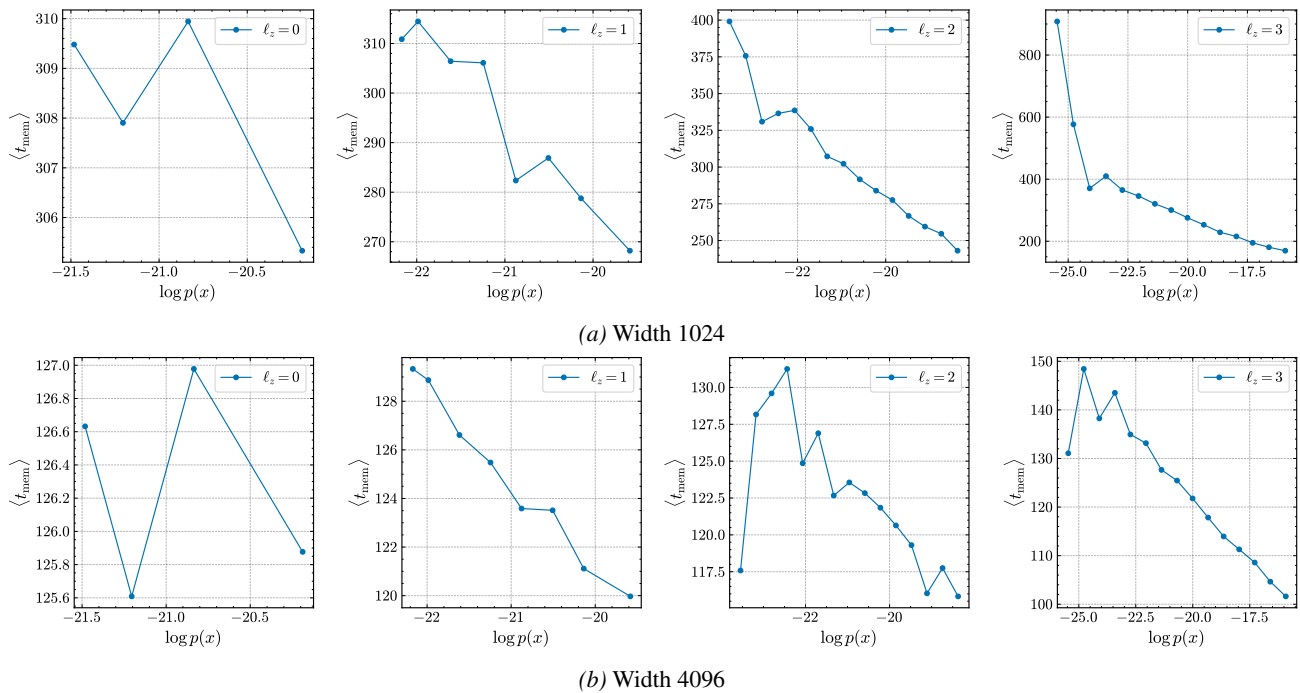

*Figure 11.* **Models memorize data points with common features earlier for $\ell_z = 1, 2, 3$.** First time $t_{\text{mem}}$ a train sample is generated, averaged across elements of similar log-likelihoods for width **a)** 1024 and **b)** 4096 models when inserting a Zipf distribution with exponent $\alpha = 1$. Figures were generated by sampling $10^5$ data points at each checkpoint.

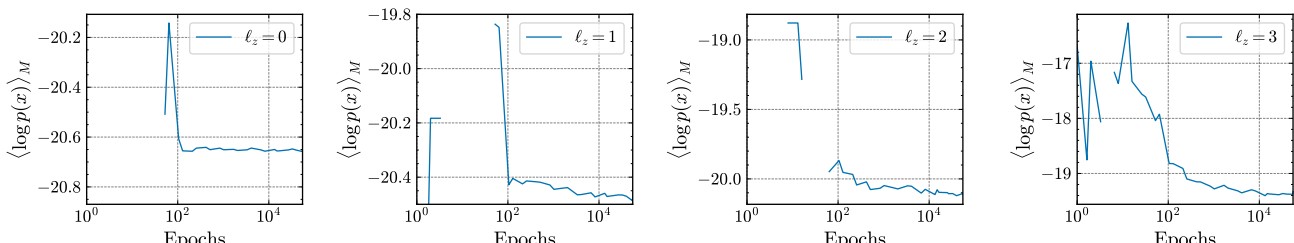

*Figure 12.* **Memorized samples are higher likelihood at early times.** At each training checkpoint of width 1024 models we generate $10^5$ data, and amongst the collection of generated samples $M$ belonging to the train set, measure the average log-likelihood. The decrease in average log-likelihood with time reflects the gradual memorization of rarer samples.

### B.3. Partial memorization

**Under-parameterized models can exhibit partial memorization.** Despite a minor increase in memorization, the evolution of the curve of valid samples in Figure 10a shows a similar behavior to that of larger-sized models: after a peak, valid samples exhibit a decrease in counts. We attribute this behavior to model repurposing of neurons in Section 4.3. In Figure 13, we can see that despite the curve for complete memorization remaining low, the model is still partially memorizing. This highlights the importance of measuring memorization for lower-level features, as complete memorization does not capture this phenomenon. Interestingly, while all models preferentially memorize the most frequent rule, this effect is more pronounced in the width 256 model, relative to the remaining rules.

### B.4. Distribution of samples

**Smaller models exhibit a longer divergence window from the target distribution.** Figure 14 shows that as model size decreases, the time window for which the average KL divergence between the generated and target distribution peaks becomes wider. This highlights a key challenge, as although smaller models are often selected to reduce memorization, their memorization dynamics can be less apparent, whilst at the same time increasing the bias toward common features. Determining the appropriate time at which to halt training therefore remains crucial to ensure correct performance.

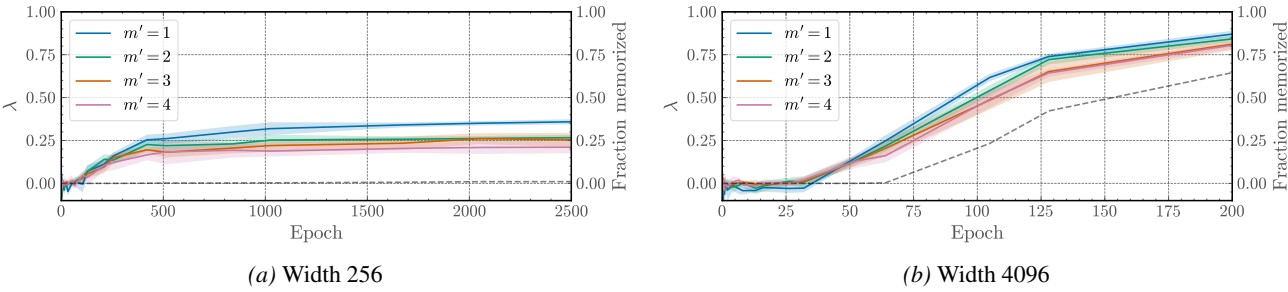

*(a)* Width 256        *(b)* Width 4096

*Figure 13.* **Partial memorization decreases with model size.** Evolution of the estimator $\lambda$ for partial memorization for width **a)** 256 and **b)** 4096 models, averaged across rules. The dashed curves show the evolution of complete memorization for either model.

*(a)* Width 256

*(b)* Width 1024

*(c)* Width 4096

*Figure 14.* **Model overproduction of common features occurs for different model sizes.** Evolution of KL divergence for model widths **a)** 256, **b)** 1024 and **c)** 4096 with Zipf exponent $\alpha = 1$. We include the subtuple count evolution at the Zipf-inserted level for models $\ell_z = 1, 2, 3$.

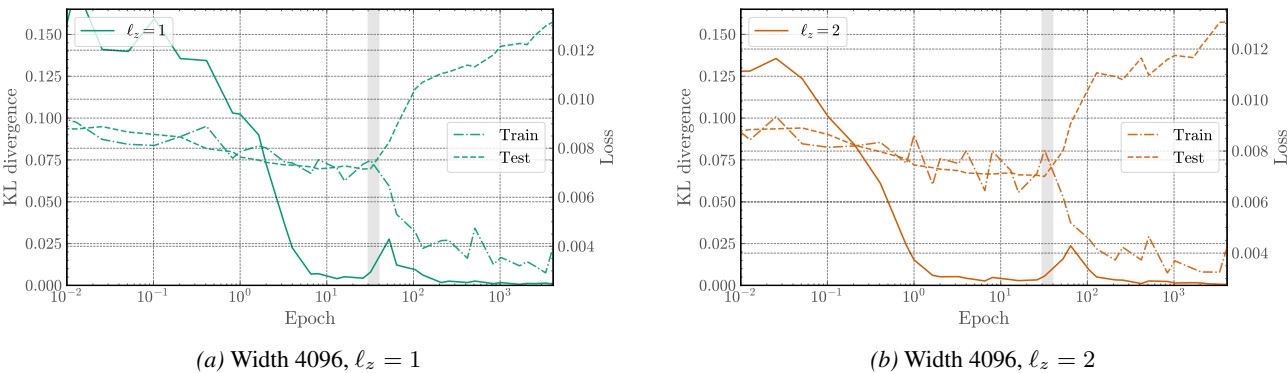

*(a)* Width 4096, $\ell_z = 1$          *(b)* Width 4096, $\ell_z = 2$

*Figure 15.* **The divergence of train and test loss may overlap with the initial increase in KL divergence**

## C. Supplementary derivations

### C.1. Dependence of $\langle t_{\mathrm{mem}} \rangle$ on $\alpha$

Figure 3 and Figure 11 suggest that $t_{\mathrm{mem}}(x) \approx \beta \log(L(x)) + c$, with $\beta < 0$ that depends on model width and $\ell_z$. Fixing width and $\ell_z$, we can compute the dependence of $\langle t_{\mathrm{mem}} \rangle$ on $\alpha$ via $\langle \log L(x) \rangle_D(\alpha)$.

The log-likelihood of a sample $x$, using production rules $m_i'^{(\ell)}(x)$ is given by

$$\log L(x) = \sum_{\ell=0}^{L-1} \sum_{i=1}^{s^\ell} \log p(m_i'^{(\ell)}) \tag{1}$$

The sum over $\ell \neq \ell_z$ contributes an uninteresting constant term $c_1 = -((s^L - 1)/(s-1) - s^{\ell_z}) \log m$ from the $p(m') = 1/m'$ that applies for all $\ell \neq \ell_z$. Therefore

$$\log L(x) = c_1 + \sum_{i=1}^{s^{\ell_z}} \log p(m_i'^{(\ell)}) \tag{2}$$

The $\log p(m_i'^{(\ell)})$ are i.i.d. random variables with finite variance, so $\log L(x)$ should converge to a normal distribution for large $\ell_z$, consistent with Figure 9. In expectation, we have

$$\langle \log L(x) \rangle_D = c_1 + s^{\ell_z} \langle \log p(m') \rangle . \tag{3}$$

Approximating the discrete Zipf distribution with a continuous one for $m' \in [1, m]$, we have $p(m') \approx \frac{\alpha-1}{1-m^{1-\alpha}} m'^{-\alpha}$. Therefore

$$\langle \log p(m') \rangle_D = \log \left( \frac{\alpha-1}{1-m^{1-\alpha}} \right) - \alpha \int_1^m \log(m') \frac{\alpha-1}{1-m^{1-\alpha}} m'^{-\alpha} \, dm'$$

and integration by parts yields

$$\langle \log p(m') \rangle_D = \log \left( \frac{\alpha-1}{1-m^{1-\alpha}} \right) - \frac{\alpha}{\alpha-1} + \frac{m^{1-\alpha} \log m}{1-m^{1-\alpha}} .$$

Combining this with Equation (3) yields

$$\langle \log L(x) \rangle_D(\alpha) = -((s^L - 1)/(s-1) - s^{\ell_z}) \log m + s^{\ell_z} \left( \log \left( \frac{\alpha-1}{1-m^{1-\alpha}} \right) - \frac{\alpha}{\alpha-1} + \frac{m^{1-\alpha} \log m}{1-m^{1-\alpha}} \right) , \tag{4}$$

which is an increasing function with $\alpha$, and therefore predicts that, with $\beta < 0$, $\langle t_{\mathrm{mem}} \rangle$ decreases with $\alpha$.

## D. CelebA

### D.1. Architecture

We train Denoising Diffusion Probabilistic Models (DDPMs) (Ho et al., 2020) on a subset of images of the CelebA dataset (Liu et al., 2015) of $64 \times 64$ pixels.

We use the Diffusers library in Python to build a U-Net architecture (Ronneberger et al., 2015) consisting of 5 downsampling and upsampling blocks (each containing two layers per block) with channel widths (128, 128, 256, 512, 512). The two smallest upsampling and downsampling blocks contain Attention blocks. In total, our model consists of 109M trainable parameters.

We also implement a smaller variant of the architecture with 4 downsampling and upsampling blocks with channel widths (64, 128, 256, 512). This smaller model consists of 63M trainable parameters.

### D.2. Training and sampling details

We train all diffusion models using Adam with a learning rate of $1 \times 10^{-5}$ and batchsize 512. During training, we use 1000 diffusion timesteps with DDPMs (Ho et al., 2020), and at inference time, to reduce computational cost we sample using 100 timesteps with Denoising Diffusion Implicit Models (DDIMs) (Song et al., 2021a).

We vary the size of the dataset used to train the models, with 1,000 (cf. Figure 16) to probe full memorization, and 10,000 (Figure 8) to probe better the generalization-memorization transition. Throughout training, we take checkpoints of the models and sample 50,000 images which we use to measure generalization and memorization.

In particular, the ability of a model to generalize and produce high-quality images is evaluated using the FID score (Heusel et al., 2017). We extract activations from the final average pooling layer (2048-dimensional) of a pretrained Inception-v3 model for a test set of 50,000 images from CelebA. We consider a model to have generalized once the FID score of sampled images plateaus.

### D.3. Memorization

To identify memorized images, we use the definition implemented by (Bonnaire et al., 2025). An image $\mathbf{x}$ is considered to be memorized if the following inequality holds:

$$|\mathbf{x} - \mathbf{x}_{NN}^{(1)}| < \frac{1}{3}|\mathbf{x} - \mathbf{x}_{NN}^{(2)}|,$$

where $\mathbf{x}_{NN}^{(k)}$ is the $k$th nearest neighbor of $\mathbf{x}$ in the train set.

Throughout our experiments we measure complete memorization through the fraction of memorized images generated per checkpoint, similar to Section 3.3.

Additionally, for Figure 7 and Figure 16b, we use *time to memorization* $t_{\text{mem}}$ as the earliest training checkpoint (in epochs) at which a train sample is considered memorized according to the above criterion.

### D.4. Log-likelihood estimation

To assess our central claim that likelier images are memorized first, we need a method to ascribe likelihoods to images. Each image in the training set has 40 attributes, some of which are rarer than others (e.g. rarest is "bald") and some of which have strong correlations (e.g. "red hair" and "blond hair" are strongly anti-correlated, while "bald" and "man" are strongly positively correlated).

To capture both individual feature rarity and these correlations, we use an Ising model, which gives a set of attributes $\sigma = (\sigma_1, \sigma_2, \ldots, \sigma_n)$ the log-likelihood:

$$\log(p(\mathbf{x})) \sim \sum_i \sigma_i \left( h_i + \sum_j J_{ij} \right),$$

where $\sigma_i = +1$ when attribute $i$ is present in the sample (e.g. red hair, bald, etc.) and $\sigma_i = -1$ when it is not.

The fitting parameters $h_i$ and $J_{ij}$ respectively capture the frequency of different attributes, and first order couplings between them. We fit the parameters $h_i$ and $J_{ij}$ with the pseudolikelihood estimator

$$PL(\sigma) \sim \prod_i p(\sigma_i|\sigma_{-i})$$

via logistic regression. This allows us to evaluate the rarity of a particular sample.

In practice, we exclude the attribute "wearing a necklace" due to the cropping of images when resizing to $64 \times 64$ pixels making this attribute unidentifiable. Additionally, the attribute "mouth slightly open" is removed due to inconsistencies when manually checking labels, also noted in (Wu et al., 2023). All other 38 attributes are used in the estimation of the log-likelihood.

### D.5. Classifier

Unlike train images from CelebA, generalized images that are not memorized do not have labels of their attributes. Therefore, to estimate the log-likelihood of such images, we require an additional classifier that estimates the necessary labels.

We fine-tune ConvNeXt-nano V2 (Woo et al., 2023) with multiclassification on the CelebA training set downsampled to $64 \times 64$ pixels. In particular, we fine-tune for 6 epochs using AdamW with a learning rate of $10^{-5}$ and a batchsize of 64. During the first epoch, we only train the classifier head using a learning rate of $10^{-4}$.

On the CelebA test set, we achieve an average of 90.6% accuracy across all attributes, ranging from 70.5% to 99.4%.

### D.6. Additional results

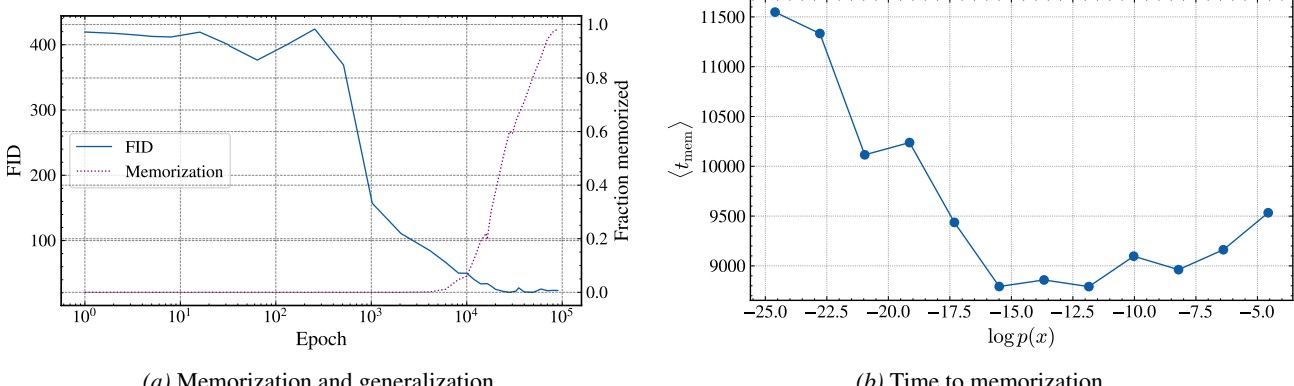

*(a)* Memorization and generalization         *(b)* Time to memorization

*Figure 16.* **Training to full memorization.** We train a model with 109M parameters on a subset of CelebA of 1,000 images. **a)** Evolution of FID (generalization) and the fraction of memorized samples. **b)** Time to memorization $t_{\mathrm{mem}}$, averaged across train images of similar estimated log-likelihood. While our observations on the RHM (Figure 3) show a purely linear pattern, here a slight uptick is observed, reflecting the increased complexity of real-world data relative to our synthetic setting.

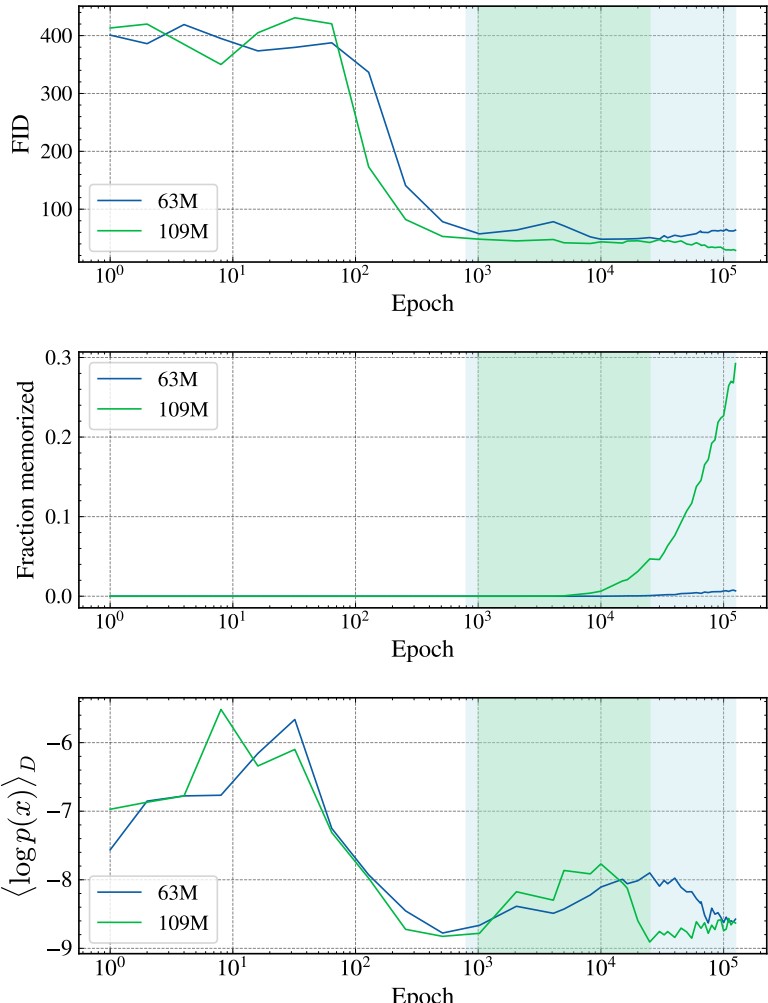

*Figure 17.* **Training dynamics on CelebA for different model sizes.** Evolution of generalization (FID), memorization and the estimated log-likelihood of images for models of size 63M and 109M parameters and trained on a dataset of 10,000 images. The grey-shaded region indicates the pre-generalization regime of both models. The blue and green shaded regions indicate the slop phase of each of the 109M and 63M models respectively, as indicated by the increase in log-likelihood. Examples of images generated by each model are presented in Figure 18 and Figure 19.

Generated Images

Train Nearest Neighbor

*(a)* Epoch 256, before generalization

Generated Images

Train Nearest Neighbor

*(b)* Epoch 1024, early generalization

Generated Images

Train Nearest Neighbor

*(c)* Epoch 10,000, "slop" phase (& early memorization)

Generated Images

Train Nearest Neighbor

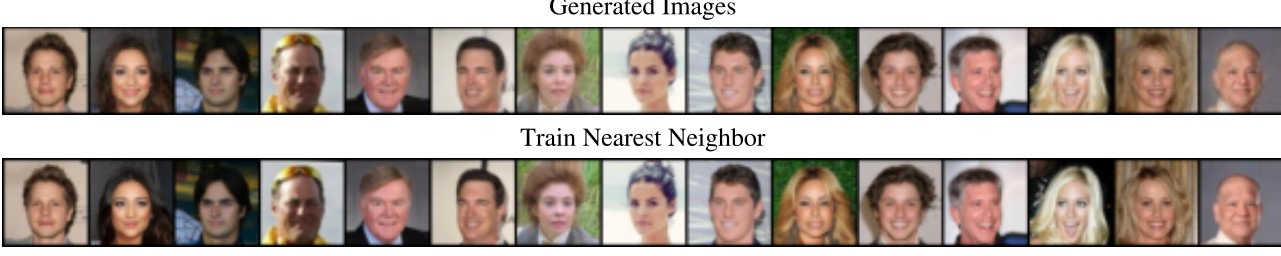

*(d)* Epoch 125,000, memorization phase

*Figure 18.* **Progression of generated images across epochs.** Generated images *(top)* and nearest neighbor *(bottom)* for the model with 109M parameters. Checkpoints shown correspond to the different stages of training: before generalization (epoch 256), beginning of generalization (epoch 1024), beginning of memorization (epoch 10,000), memorization (epoch 125,000).

Generated Images

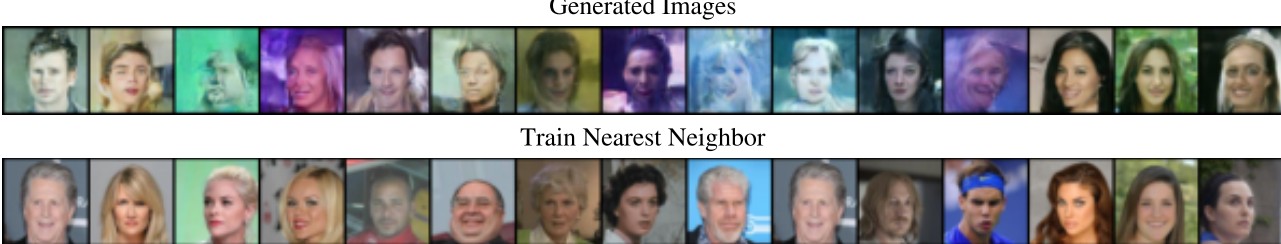

Train Nearest Neighbor

*(a)* Epoch 256, before generalization

Generated Images

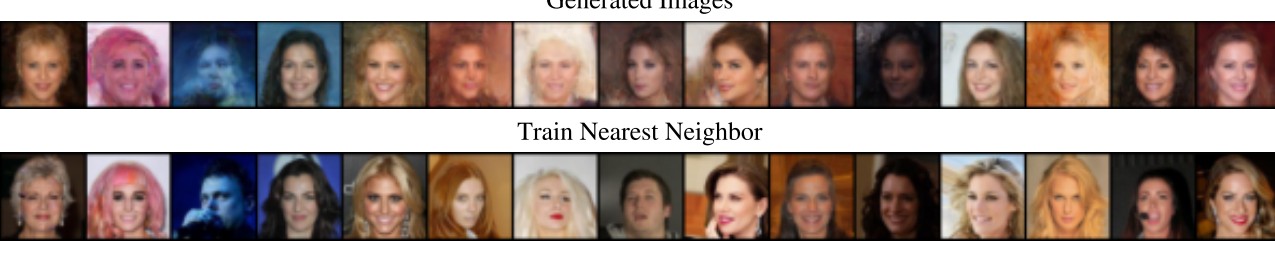

Train Nearest Neighbor

*(b)* Epoch 1024, early generalization

Generated Images

Train Nearest Neighbor

*(c)* Epoch 10,000, "slop" phase

Generated Images

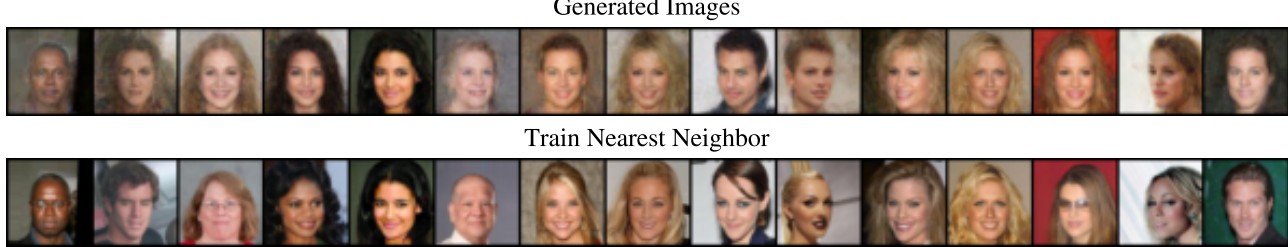

Train Nearest Neighbor

*(d)* Epoch 125,000, "slop" phase (& early memorization)

*Figure 19.* **Progression of generated images across epochs.** Generated images *(top)* and nearest neighbor *(bottom)* for the model with 63M parameters. The same checkpoints as in Figure 18 are shown for direct comparison with the 109M parameter model.

