# OpenReview forum: "Diffusion Models Preferentially Memorize Prototypical Examples or: Why Does My Diffusion Model Love Slop?"
_ICML.cc/2026/Conference — ICML 2026 regular_

### Official Review · Reviewer_QMDc · 2026-03-02

**Soundness:** 3
**Presentation:** 3
**Significance:** 3
**Originality:** 3
**Overall Recommendation:** 5
**Confidence:** 4

**Summary:**

In this manuscript, memorization in Diffusion models is studied through the lens of the Random Hierarchy Model (RHM). The RHM defines a distribution of data with high-level and low-level features, organized by a set of hierarchical production rules which assigns a probability to observing a particular pattern of data which does not necessarily encompass the whole data dimension. This allows the authors to define partial memorization, a phenomenon where an excess of particular patterns are generated by the model, driven by the presence of those patterns in the training data.

**Compliance With Llm Reviewing Policy:**

Affirmed.

**Final Justification:**

The authors answered all my questions, I maintain my "accept" score.

**Key Questions For Authors:**

4) Figure 1 presents a series of hypotheses on how memorization occurs in diffusion models, but is not referred to beyond the introduction. I  think that the presentation of the paper could be improved by also discussing which of the competing hypotheses in Figure 1 can be rejected by the experiments either in the results or discussion section.

5) Figure 3: Does partial memorization occur before or after Valid samples are produced? The choice of axes makes it hard to tell.

**Limitations:**

yes

**Strengths And Weaknesses:**

Strengths:

1) The authors approach the question of memorization in a more fine-grained manner than before, namely by asking if particular sub-patterns are preferentially generated by a diffusion model, reflecting a bias in the training data.

2) The authors provide a series of interesting experiments on the basis of RHM data, such as showing that prototypical patterns are memorized first, and that the partial memorization occurs earlier in training than full memorization.

Weaknesses

3) The most important point is surely that the results rely only on the RHM, a synthetic model of data. While the RHM has reproduced behaviors that are also observed in diffusion models trained on real data, I find it important that this is also shown in this case, at least qualitatively, or that the claims are adapted accordingly. For example, the result that prototypical patterns are memorized first could be demonstrated e.g. in a real data set, by examining the distributions of features of image classifiers on real data and on generated data, or by another method that the authors deem suitable.

---

> ### Author Rebuttal · Authors · 2026-03-31
>
> We thank Reviewer QMDc for their positive review and thoughtful comments. We address the questions raised below:
>
> ---
> **Experiments on real data**
>
> We agree with the reviewer that including additional experiments to bridge the gap between our synthetic setting and real data is valuable. To do this, we conducted additional experiments on CelebA.
>
>
> While CelebA lacks the hierarchical latents we have access to in the RHM, it provides labels of 40 attributes for each image in the dataset. We used these attributes to estimate (using an Ising model) the likelihood of each image. We then replicated the RHM experiments, confirming that data composed of common features is memorized first. We also showed that models exhibit a shift towards generating images with higher likelihood (containing more common features).
>
>
> To do so, we trained a DDPM with a Unet of 108M parameters on a subset of CelebA, ensuring we observe a phase of generalization before memorization occurs. We measure memorization using the nearest neighbours criterion from [1]. Throughout training, we take checkpoints of the model and sample 40000 images at each checkpoint, which we use to evaluate memorization.
>
> **(i)** By using the likelihood estimated through the Ising model, we consider the earliest time train elements are reproduced, and use this to produce a plot as in Figure 3b. Our results show that lower likelihood images on average are memorized later in training.
>
>  **(ii)** We finetune ConvNeXt-nano v2 (a strong pretrained classifier) to identify the same 40 attributes present in images generated by the diffusion model, and, in turn, use this to evaluate the average log-likelihood of images generated at each checkpoint (including non-memorized images). We observe that memorization is preceded by an initial increase in the average log-likelihood of samples generated, which continues increasing as the model memorizes. This suggests a shift in the generated distribution towards more common features, similar to the observations in Figure 6 for the RHM.
>
> Our results can be found in the following anonymous link: https://anonymous.4open.science/r/icml_rebuttal_2026-0926/
>
> We have provided complete details of the experiments in the response to Reviewer xjvJ (experiments on larger image datasets). Our results confirm that the observations from our experiments on the Random Hierarchical Model (RHM) generalize beyond synthetic data, reinforcing the validity of our conclusions.
>
> [1] Bonnaire, T., Urfin, R., Biroli, G., and Mezard, M. Why diffusion models don’t memorize: The role of implicit dynamical regularization in training.
>
> ---
> **Reference to hypotheses in Figure 1**
>
> We agree that this can be made clearer. Our experiments in Section 4.1 resolve the first set of hypotheses (first row of Figure 1), and the results in Section 4.2 provide evidence for the second set. We will revise the text to explicitly connect these results to the corresponding hypotheses in Figure 1, both in the main sections and in the conclusion.
>
> ---
> **Partial memorization and generalization in Figure 3**
>
> Partial memorization occurs well after the first valid samples are produced. The model produces valid samples with 50% probability at around epoch 10, while the initial increase in the estimated $\lambda$ occurs around Epochs 50-60.

---

> > ### Author Rebuttal · Reviewer_QMDc · 2026-04-01
> >
> > I thank the authors for answering my questions and for providing further experiments. As I have already given a good rating, I will not raise my score.

---

### Official Review · Reviewer_dKgU · 2026-03-08

**Soundness:** 3
**Presentation:** 3
**Significance:** 3
**Originality:** 2
**Overall Recommendation:** 4
**Confidence:** 4

**Summary:**

This paper investigates the memorization dynamics of diffusion models to determine whether they preferentially memorize rare outliers or common prototypical examples, utilizing 1D U-Nets trained on hierarchically structured discrete data generated by the Random Hierarchy Model (RHM). To precisely track this phenomenon, the authors introduce a "partial memorization" estimator ($\lambda$) that mathematically decouples legitimate statistical generalization from the rote memorization of specific high-frequency sub-features. The empirical results demonstrate that models do not memorize rare outliers first. Instead, they undergo a partial memorization phase where they preferentially memorize and overproduce common, prototypical sub-features, which is a behavior the authors liken to the generation of the "slop". Furthermore, the study finds that fat-tailed dataset distributions successfully delay the onset of memorization, with the defensive effect being most prominent when variance is introduced at higher, more abstract levels of the data hierarchy.

**Compliance With Llm Reviewing Policy:**

Affirmed.

**Final Justification:**

The paper presents an study on memorization dynamics in diffusion models. I appreciated the original $\lambda$ metric but had major reservations regarding its applicability to real-world data and potential sampling artifacts.
The newly added CelebA experiments successfully demonstrate that the observed memorization dynamics hold in natural, entangled data. Scaling the sampling size from $10^5$ to $10^6$ ruled out my concerns about statistical sampling bias.
The rebuttal fully resolved my core concerns, so I am raising my score to **4: Weak accept**

**Key Questions For Authors:**

1. In Section 4.3, the authors claim that valid samples drop because "neurons are reallocated" to memorize training data. Is there an analysis of the model's weights, activations, or internal layers?
2. The memorization time ($t_{mem}$) is measured using a test batch of 10,000 generated samples, whereas the training set has 20,000 samples. Because rare samples naturally have a lower probability of being generated, they might appear later in the test batches simply by chance, even if the model has already memorized them. This makes it unclear whether the delayed $t_{mem}$ is a true learning effect or just a statistical sampling delay.

**Limitations:**

Yes.

**Strengths And Weaknesses:**

**Strengths:**
1. **Clear writing and structure:**
The authors clearly compare three different hypotheses about what models memorize first. They also use helpful figures (like Figure 2) to simply explain their complex synthetic dataset. This clear presentation helps readers easily understand the core problem.
2. **Clean experimental design:**
The experimental design is clean and highly controlled. The authors make sure every training sample is completely unique to remove the effect of exact data duplication. This strict setup clearly proves that models memorize common sub-features early, rather than just repeating whole samples. The experimental results build a strong and logical chain of evidence from start to finish.
3. **The $\lambda$ metric for partial memorization:**
The main original contribution is the new lambda ($\lambda$) metric for measuring "partial memorization". The authors notice a tricky problem: a good model will naturally generate common features just by learning the rules correctly. To fix this, the $\lambda$ metric mathematically calculates and removes this expected natural overlap. This provides a solid and accurate way to separate true feature-level copying from normal learning.

**Weakness:**
1. **The gap between toy data and real entangled data:**
The paper uses a handcrafted dataset. In this synthetic data, features are perfectly separated into clean levels. Whereas real features are deeply mixed and entangled. Because of this huge gap, we cannot be sure these toy results apply to complex real-world models.
2. **The practical advice is impossible to follow:**
The authors use their results to give practical advice. For example, they suggest "careful checkpoint selection" to avoid generating "slop" (in Section 4.3 and Section 5). However, they only found this "slop" problem by using the $\lambda$ metric. We cannot calculate $\lambda$ without knowing the exact hidden rules of the data. In the real world, we never know these exact rules. Therefore, practitioners cannot actually measure this problem or follow the authors' advice.

---

> ### Author Rebuttal · Authors · 2026-03-31
>
> We appreciate Reviewer dKgU’s detailed feedback. We respond to the points mentioned in the review below:
>
> ---
> **Experiments on real data**
>
> The RHM model allows exhaustive evaluation that would be hard to replicate completely on natural data. We do agree experiments on natural data would strengthen the results. To do this, we conducted additional experiments on CelebA.
>
> While CelebA lacks the hierarchical latents we have full access to in the RHM, it does provide labels of 40 attributes for each image in the dataset. We used these attributes to estimate (using an Ising model) the likelihood of each image. We then trained a DDPM with a Unet of 108M parameters on a subset of CelebA, ensuring we observe a phase of generalization before memorization occurs. We measure memorization using the nearest neighbours criterion from [1]. Throughout training, we take checkpoints of the model and sample 40000 images at each checkpoint, which we use to evaluate memorization.
>
> **(i)** By using the likelihood estimated through the Ising model, we consider the earliest time train elements are reproduced, and use this to produce a plot as in Figure 3b. Our results show that lower likelihood images on average are memorized later in training.
>
>  **(ii)** We finetune ConvNeXt-nano v2 (a strong pretrained classifier) to identify the same 40 attributes present in images generated by the diffusion model, and, in turn, use this to evaluate the average log-likelihood of images generated at each checkpoint (including non-memorized images). We observe that memorization is preceded by an initial increase in the average log-likelihood of samples generated, which continues increasing as the model memorizes. This suggests a shift in the generated distribution towards more common features, similar to the observations in Figure 6 for the RHM.
>
> Our results can be found in the following anonymous link: https://anonymous.4open.science/r/icml_rebuttal_2026-0926/
>
> The full details of the experiments can be found in the response to Reviewer xjvJ (experiments on larger image datasets).
>
> [1] Bonnaire, T., Urfin, R., Biroli, G., and Mezard, M. Why diffusion models don’t memorize: The role of implicit dynamical regularization in training.
>
> ---
> **Practical advice**
>
> The reviewer’s concern is that our analysis relies on access to the true latent structure of the data. While this is available in the RHM, our rebuttal to xjvJ shows that it is not required in practice. For natural image data we introduce a pseudolikelihood-based metric defined over observable semantic attributes (e.g., facial features), which does not depend on knowledge of the underlying generative process. This proxy is sufficient to surface the same failure mode.
>
> More importantly, beyond its interpretation as practical guidance, our work identifies and characterizes a previously unformalized regime of partial memorization and over-replication. While such behavior is widely observed in practice (e.g., generative models reproducing common patterns), there has been limited understanding of when and why it arises. Our analysis provides a concrete mechanism and diagnostic for this phenomenon.
>
> ---
> **Neuron repurposing**
>
> We attribute this to neuron ‘repurposing’, because the drop in performance is partially ameliorated by increasing model size (cf. Fig. 8). If the network width is much greater than the number of neurons necessary for generalization, then partial memorization can be accomplished without harming the generation of valid samples. We will revise the text to explain our thinking in this regard.
>
> ---
> **Sampling size**
>
> We have increased the sample size from $10^5$ to $10^6$ and resampled with a new random seed to improve statistics and reproduce figures 3b and 9. We observe the same pattern in all models, where higher log-likelihood data appears earlier. We will update Figure 3b and Figure 9.

---

> > ### Author Rebuttal · Reviewer_dKgU · 2026-04-03
> >
> > Thank the authors for the comprehensive rebuttal and the new CelebA experiments, which resolve my concerns regarding real-world applicability and sampling bias. I am raising my score to **4: Weak accept**, and I have no further questions.

---

### Official Review · Reviewer_EF8b · 2026-03-13

**Soundness:** 3
**Presentation:** 3
**Significance:** 3
**Originality:** 3
**Overall Recommendation:** 4
**Confidence:** 3

**Summary:**

This paper studies which training samples get memorized first by diffusion models. The authors use RHM, a synthetic data framework with explicit hierarchical compositional structure, to run controlled experiments on memorization dynamics. The core finding is that samples composed of common subfeatures are memorized before rare ones, even when the dataset is fully deduplicated. The paper also introduces a formal notion of partial memorization, shows that fat-tailed datasets delay overall memorization, and connects this to the slop phenomenon where early-stopped models still over-generate bland, prototypical outputs.

**Compliance With Llm Reviewing Policy:**

Affirmed.

**Key Questions For Authors:**

-	Is there any preliminary evidence, even qualitative, that the common-first memorization ordering holds for real datasets like CIFAR or a small text corpus? Even a simple plot of memorized samples versus their estimated frequency would help bridge the gap between the RHM setting and the claimed implications.
-	Why do you think the preference for memorizing common data points disappears when Zipf's law is at the root level? This seems to contradict the main story, and the paper does not adequately address it.
-	How robust is the lambda estimator to violations of the mixture model assumption? Specifically, what happens if the model learns a non-uniform distribution over valid tuples that is not the training distribution? Would this produce spurious lambda > 0 values?
-	Can you give a more precise definition of what counts as slop in your framework? Is there a metric on generated output quality or diversity that you could compute to demonstrate that early-stopped models in the partial memorization regime produce outputs that are measurably less diverse or creative?

**Limitations:**

Yes

**Strengths And Weaknesses:**

Strengths

-	The research question is well-motivated and timely. The three competing hypotheses are clearly stated, and the paper gives a clean answer to which one holds for diffusion models.
-	The use of RHM is a genuinely good choice. It gives exact ground truth about sample likelihood and compositional structure, allowing exhaustive evaluation that would be impossible on natural data.
-	The partial memorization estimator is a nice contribution. Disentangling fair sampling from memorization bias through a mixture model is principled and well-executed.
-	The connection to slop is clever and adds real-world relevance to what could otherwise be a dry mechanistic study.
-	Experiments are thorough across model sizes and across levels of the hierarchy. The figures are clear and tell a coherent story.

Weaknesses

-	The RHM is far from natural data, and the generalization gap is not bridged. The RHM has a very specific structure: production rules are random, fixed, and have clean power-law distributions by construction. Real image or text datasets do not have this clean factored compositional structure. The paper never attempts even a qualitative validation of its main claims on natural data. For instance, do images containing common textures or objects get memorized earlier in practice? Do text sequences with frequent n-grams appear earlier in memorized outputs? Without any such check, the gap between the controlled synthetic setting and the claimed broader implications remains wide. The paper discusses this in the limitations but does not make any attempt to close it, even informally.
-	The partial memorization metric has a fragile assumption. The estimator in Definition 3.2 assumes that the model either copies a tuple from the training set with probability lambda, or samples uniformly at random from valid tuples. This mixture model is a simplification. In reality, a model could be biased toward certain tuples without outright copying them, which would not be captured by this estimator. The paper does not discuss how sensitive the results are to this assumption, nor whether the estimator is consistent under model misspecification.
-	The $\mathcal{l}=0$ case is unexplained. Figure 9 and Appendix B.2 show that when Zipf's law is introduced at the root level ($\mathcal{l}=0$), the common-first memorization ordering does not appear. The paper notes this as "interesting" but provides no real explanation. This is actually a puzzling result that cuts against the main narrative of the paper. If the mechanism were purely about gradient reinforcement from common sub-features, it should apply at all levels. The failure at $\mathcal{l}=0$  suggests the mechanism is more subtle, and this deserves more attention.

Minor Weaknesses:
-	Only one value of alpha (α = 1) is used in most figures. More alpha values in the core experiments would strengthen claims about the generality of the effects.
-	The conjecture in Section 4.2 about why class-level variation delays memorization (“the model sees proportionally more examples of rare features at lower levels”) is reasonable but not tested directly.

---

> ### Author Rebuttal · Authors · 2026-03-31
>
> We thank Reviewer EF8b for their useful comments and thorough review. We address below the questions raised in the review:
>
> ---
> **Experiments on real data**
>
> We agree that our conclusions would be strengthened by validating our hypotheses in natural data. We now do this for image diffusion models trained on CelebA, and for space reasons refer the referee to our response to referee xjvJ for full details.
>
> To summarize the new results: (1) The new experiments confirm that diffusion models preferentially memorize images with higher likelihood (corresponding to Fig 3b on the RHM).  (2) We confirm the existence of a “sloppy” output phase preceding memorization (cf. Fig. 6) where the model has begun to output data with higher likelihoods, but not regurgitate exact copies of training data.
>
> ---
> **Behaviour when Zipf’s law is introduced at root level**
>
> We observe that the effect on memorization time is weaker for variation introduced at higher levels of abstraction, hence it's possible that the strength of this effect on the root node is below our detection threshold (set by checkpoint frequency, model size, etc.). If the effect on time to memorization is genuinely zero for variation imposed at the root node, it potentially hints at something interesting in the learning dynamics of hierarchically organized data. The current line of thinking for diffusion models of the RHM is that they learn the score via token-token correlations [1]. But, if this reasoning applies also to latents, it suggests that latent-latent correlations at higher levels of abstraction are also utilized. The only latent which lacks such correlations is the root node: it is singular in the RHM. We will add this to the discussion and note this as a direction for future study in the paper.
>
> [1] Sclocchi, A., Favero, A., Itzhak Levi, N., and Wyart, M. Probing the latent hierarchical structure of data via diffusion models.
>
> ---
> **$\lambda$ assumptions**
>
> In brief, $\lambda$ measures the overproduction of training points by a generative process vis-a-vis the ground-truth sampler. It is not specific to memorization for a mixture model. Generative models that have a bias towards some decomposition rules, but which don't emit more training points than a fair sampler will still have $\lambda = 0$. This can be made evident by considering a general sampler, for which $\lambda$ can be expressed as:
>
> $$ \lambda = \frac{p_{sampler}(x \in X_{train}) - p_{RHM}(x \in X_{train})}{1-p_{RHM}((x \in X_{train}))} $$
>
> Consequently, any over-production of training points counts towards $\lambda$; we view this conservatism as a positive: any train set reproduction unexplained by the ground truth generative process should be regarded with suspicion.
>
> We will include this discussion and general formulation of $\lambda$ in the text.
>
> ---
> **Slop definition**
>
> Here, we define “slop” as generated outputs that overutilize the more common features. In our setting, we find that our models are “sloppy” at the early stages of memorization: they still can generate unseen outputs, but there is a distributional shift in those outputs towards overrepresenting common features at an abstract level (cf. Fig. 6). Recent work in the language domain highlights the excess use of  particular syntactic templates (i.e. relatively abstract compositions of text) by LLMs and identifies it with “slop”, which we view as a promising direction [1].  We will revise the text to make this definition clear and expand on the associated discussion.
>
> [1] Detection and Measurement of Syntactic Templates in Generated Text  Chantal Shaib, Yanai Elazar, Junyi Jessy Li, Byron C Wallace
>
> ---
> **Experiments using a wider range of $\alpha$ values**
>
> We sweep across variation in $\alpha$ in Fig. 4 with the purpose of testing (and validating) the hypothesis that broad distributions are memorized more slowly. Such simulations can also serve as an independent test of our $\lambda$ measure (i.e. corresponding to Fig. 3c), and we will include these measurements in the revised appendix. Although one could run experiments on a wider RHM hyperparameter grid (model width, alpha, Zipf level), such experiments would mostly refine predictions and patterns already observed in the synthetic setting. Instead, we focused on directly running experiments on natural data to test real-world applicability of the theory, which we view as the more meaningful validation.
>
> ---
> **Hypothesis in Section 4.2 about class-level variation delaying memorization**
> The reviewer raises an interesting point. The conjecture in Section 4.2 is not directly tested, and is intended as an interpretation of the observed results in Figure 5. The testing of this hypothesis would require changing the generative process, e.g. to fix the likelihood distribution induced by different $\ell_z$. A careful study doing so would be a valuable contribution, but lies beyond the scope of our current paper, so we will add a note on this in the Conclusion section.

---

> > ### Author Rebuttal · Reviewer_EF8b · 2026-04-04
> >
> > I am happy with the responses, but still some weaknesses remain. I have already given this paper a score of Weak Accept and I will keep that.

---

### Official Review · Reviewer_xjvJ · 2026-03-13

**Soundness:** 2
**Presentation:** 3
**Significance:** 2
**Originality:** 3
**Overall Recommendation:** 4
**Confidence:** 3

**Summary:**

This paper studies the memorization of diffusion models in the level of features & components. Though experiments conducted on a specifically synthesized dataset, it experimentally shows that diffusion models tend to firstly memorize features that frequently appear in the dataset.

**Compliance With Llm Reviewing Policy:**

Affirmed.

**Final Justification:**

I believe the main body of my concern has been addressed by a large-scale experiment on celeba.

**Key Questions For Authors:**

1, Is it possible to migrate your experiments to larger image datasets like CelebA, since on facial data we can also observe clear features, both frequent ones and rare ones? I would love to raise my score if you can show me results on larger datasets.

**Limitations:**

yes

**Strengths And Weaknesses:**

Strengths:

[1] This paper focuses the memorization of diffusion models to the feature level and studies partial memorization for the first time. This is novel to the best of my knowledge.

[2] Evidence and analysis are both sufficient on the synthesized dataset to support the hypothesis proposed that diffusion models do preferentially memorize frequently appearing features, at least on that synthesized dataset.
Weaknesses:

The only and outstanding weakness of this paper is its experimental setting. Based on my understanding, the experiment is only conducted on a small-scale (of 20000 samples) synthesized discrete dataset, consisting of compositions of discrete features, and the author uses Gaussian diffusion to fit these discrete samples. This is not a persuasive and generalized setup as far as I am concerned for the following two reasons:

First, the dynamic of diffusion models could dramatically vary based on the composition of parameterizations (Gaussian & discrete) and the dataset. This means that we cannot simply generalize the conclusion of one combo to others. Hence, to study the memorization of the basic setting of Gaussian diffusion, one should at least conduct experiments on simple image datasets like Swiss Roll, rather than highly dispersed discrete data points. Furthermore, the composition of Gaussian diffusion and discrete data must have a different optimal noise schedule and other setups. One cannot simply use the diffusion setups tuned on image data to fit discrete data.

Second, the experiment scale is very limited so that some important factors may not be well considered. For example, the size of a model used to be considered as a critical factor to the memorization of diffusion models. However, this paper only conducts experiments on two sizes of models, which raises questioning whether the conclusion is generalizable.

-------------
Update: I have adjusted my score to 4 after the rebuttal.

---

> ### Author Rebuttal · Authors · 2026-03-31
>
> We thank Reviewer xjvJ for recognizing the novelty of our approach and providing constructive feedback. We address the question raised below.
>
> **Experiments on larger image datasets**
>
> Although the synthetic setup is necessary to provide a controlled and interpretable setting and allows analysing the latents at multiple layers of abstraction, the gap between synthetic and real data can be more closely bridged, and we have conducted additional experiments. In brief, using a diffusion model trained on celebA and a sample-likelihood estimator, we find that images composed of more common features are memorized first, validating our findings from the synthetic setting. We will include these new experiments in the main text, as they considerably strengthen the paper.
>
> This additional experimental setting consists of four steps:
> (i) We follow the reviewer’s advice and train DDPMs with 108M parameters on a subset of CelebA images that generalize at early times before memorizing at later times.
> (ii) We identify memorized images following the method of [1].
> (iii) We ascribe to images a log-likelihood based on their attributes, using an Ising-model based estimator.
> (iv) We make the following measurements:
> - The time to memorization as a function of sample likelihood. We replicate the observations made in the synthetic setting, that diffusion models memorize samples composed of common features at earlier times (i.e. the results of Fig. 3b and 9).
> - We measure the mean log-likelihood of generated samples as a function of training time. We find that just as in the synthetic setting (cf. fig. 6), memorization is preceded by an increase in sample log-likelihood, i.e. models produce outputs composed of more frequent combinations of features.
>
> Results from these experiments can be found at the following anonymous link: https://anonymous.4open.science/r/icml_rebuttal_2026-0926/
>
> A full set of validation experiments for the multiclassifier will be included in the appendix of the updated paper, along with details related to the fitting procedure of the Ising model estimator and its consistency with the test set.
>
> Details on setup:
>
> (i) We train DDPM models using the standard diffusers library with 108M parameters. As the number of training steps and model parameters necessary to observe memorization scale with dataset size, we choose a random subset of the CelebA training set consisting of 1000 labelled images (64x64 pixels), consistent with [1]. This is a sufficient number of images necessary to achieve generalization at early times, as we verify. We take checkpoints during training and sample 40000 images at each. We are additionally replicating this in a larger dataset of 10000 images.
>
> (ii) To identify memorized images we use the condition
> $|x-x^1_N| < ⅓ |x - x^2_N|$,
> where $x^k_N$ is the $k$th nearest neighbour in the train set to the generated image x; as implemented in [1].
>
> (iii) To assess our central claim that likelier images are memorized first, we need a method to ascribe likelihoods to images. Each image in the training set has 40 attributes, some of which are rarer than others (e.g. rarest is “bald”) and some of which have strong correlations (e.g. “red hair” and “blonde hair” are strongly anti-correlated, while “bald” and “man” are strongly positively correlated). To capture both individual feature rarity and these correlations, we use an Ising model, which gives a set of attributes $\sigma$ the log-likelihood:
>
> $$ \log(P(\sigma)) \sim \sum_i \sigma_i ( h_i + \sum_j J_{ij} ) $$
>
> where $\sigma_i = +1$ when attribute $i$ is present in the sample (e.g. red hair, bald, etc.) and $-1$ when it is not. The fitting parameters $h_i$ and $J_{ij}$ respectively capture the frequency of different attributes, and first order couplings between them. We fit the parameters $h_i$ and $J_{ij}$ with the pseudolikelihood estimator
>
> $$ PL(\sigma) \sim \prod_i P(\sigma_i | \sigma_{-i}) $$
>
> via logistic regression. This allows us to evaluate the rarity of a particular sample.
>
> (iv) We conduct the following two tests of our theory: (a) we measure time to memorization for different memorized samples, finding that likelier samples are memorized earlier, and (b) we measure the distribution of output log-likelihoods during training, and find that after generalization, but before memorization, is a phase of increased output attribute log-likelihood. To obtain the sample attributes of output images necessary to assess their likelihood, we finetune ConvNeXt-nano v2 [2] with multiclassification on the CelebA training set (finding a mean accuracy of ~91%, ranging from 70-99% for individual attributes).
>
> [1] Bonnaire, T., Urfin, R., Biroli, G., and Mezard, M. Why diffusion models don’t memorize: The role of implicit dynamical regularization in training.
> [2] Sanghyun Woo, Shoubhik Debnath, Ronghang Hu, Xinlei Chen, Zhuang Liu, In So Kweon and Saining Xie. ConvNeXt V2: Co-designing and Scaling ConvNets with Masked Autoencoders

---

> > ### Author Rebuttal · Reviewer_xjvJ · 2026-04-03
> >
> > The author complements the weakness in their experiments, where my concern mainly locates at.

---

### Decision · Program_Chairs · 2026-04-30

**Decision:**

Accept (regular)

**Comment:**

I agree with the reviewers that this work carefully studies an important and timely problem with controlled experiments, and that non-synthetic experiments would strengthen the paper significantly. The additional experiments in the discussion period on CelebA strengthen the paper, but ideally there would more evaluation on such datasets, given the arguably very strong assumptions of RHM. These are not technical flaws, but would greatly improve the presentation, as the claims are arguably a bit too broadly scoped given such assumptions.